# Genome-wide analysis in the mouse embryo reveals the importance of DNA methylation for transcription integrity

Thomas Dahlet [1,2], Andrea Argüeso Lleida[1,2], Hala Al Adhami [1,2], Michael Dumas[1,2], Ambre Bender [1,2], Richard P. Ngondo [1,2,3], Manon Tanguy[1,2], Judith Vallet[1,2], Ghislain Auclair[1,2], Anaïs F. Bardet [1,2] & Michael Weber [1,2 ✉]

Mouse embryos acquire global DNA methylation of their genome during implantation. However the exact roles of DNA methyltransferases (DNMTs) in embryos have not been studied comprehensively. Here we systematically analyze the consequences of genetic inactivation of *Dnmt1*, *Dnmt3a* and *Dnmt3b* on the methylome and transcriptome of mouse embryos. We find a strict division of function between DNMT1, responsible for maintenance methylation, and DNMT3A/B, solely responsible for methylation acquisition in development. By analyzing severely hypomethylated embryos, we uncover multiple functions of DNA methylation that is used as a mechanism of repression for a panel of genes including not only imprinted and germline genes, but also lineage-committed genes and 2-cell genes. DNA methylation also suppresses multiple retrotransposons and illegitimate transcripts from cryptic promoters in transposons and gene bodies. Our work provides a thorough analysis of the roles of DNA methyltransferases and the importance of DNA methylation for transcriptome integrity in mammalian embryos.

[1] University of Strasbourg, Strasbourg, France. [2] Biotechnology and Cell Signaling, CNRS UMR7242, 300 Bd Sébastien Brant, 67412 Illkirch, Cedex, France. [3] Present address: IBMP, CNRS UPR2357, 67084 Strasbourg, France. ✉email: michael.weber@unistra.fr

DNA methylation in vertebrate genomes has important functions in gene regulation, development, and diseases[1]. This modification occurs at CpGs and is abundant genome-wide, except at CpG islands (CGIs) that are refractory to DNA methylation. The mouse genome encodes four active DNA methyltransferases (DNMT): DNMT1, DNMT3A, DNMT3B, and DNMT3C. Another member of the family, DNMT3L, is catalytically inactive but stimulates the activity of the other DNMT3 enzymes. DNMT3C and DNMT3L are expressed in germ cells and their inactivation leads to impaired gametic DNA methylation and infertility[2,3]. In contrast, the inactivation of DNMT1, DNMT3A, or DNMT3B in mice leads to embryonic or postnatal lethality[4,5], which illustrates the essential role of these enzymes in development.

Global genome methylation is established after implantation of the embryo in the mouse and subsequently maintained in most cell lineages. DNMT3A and DNMT3B are thought to perform all de novo methylation of DNA during development[4,6], however the consequences of the double knockout (DKO) of Dnmt3a/b on the methylome have not been investigated genome-wide. Moreover, the single inactivation of Dnmt3a or Dnmt3b has only a moderate impact on DNA methylation levels in mouse embryos[4,7], suggesting either strong redundancy or involvement of other enzymes in de novo methylation. In contrast, DNMT1 is thought to be the main enzyme responsible for maintenance DNA methylation after replication. However, DNMT1 also shows capabilities for de novo DNA methylation in vitro, in mouse embryonic stem (ES) cells, and oocytes[8–12]. Conversely, late passage Dnmt3a/b knockout ES cells show reduced methylation genome-wide[13–15] and at imprinted differentially methylated regions (DMRs)[16], suggesting that DNMT3A/B are also required for the faithful maintenance of CpG methylation in development. Despite these studies suggesting complex functions of DNMTs, the in vivo roles of these enzymes in embryonic development remain elusive. Previous investigations of the roles of DNMTs in embryos were limited to locus-specific analysis[4,6,17–19], which highlights the necessity for complete methylomes of Dnmt mutant embryos to validate models of DNMT functions in vivo.

Further work is also needed to illuminate the transcriptional roles of DNA methylation in development. Previous studies in Dnmt knockout embryos showed that DNA methylation is required to repress imprinted genes[20,21], Rhox genes[6], germline genes[22], and intracisternal A-particle (IAP) transposons[17]. However, no genome-wide transcriptome analysis in strongly hypomethylated embryos has been conducted. Moreover, transcriptome profiling in Dnmt triple knockout (TKO) mouse ES cells devoid of DNA methylation revealed only a minor impact on the expression of genes and transposable elements (TEs)[23,24]. One explanation is that ES cells use other mechanisms to compensate for the loss of DNA methylation, mimicking what is happening during epigenetic reprogramming in preimplantation embryos and primordial germ cells[25]. Indeed, the repression of endogenous retroviruses (ERVs) in mESCs is primarily mediated by KAP1 and SETDB1 responsible for H3K9me3, rather than DNA methylation[23,26–28]. In contrast, IAP repression becomes dependent on DNA methylation in differentiated cells[29], supporting the model that DNA methylation is not important for initial repression in early embryonic cells but for the transition to long-term silencing.

Here we perform a comprehensive investigation of the role of DNMTs during global genome remethylation in the mouse embryo. We report genome-wide methylomes in Dnmt1 knockout and Dnmt3a/b DKO embryos (embryonic day 8.5), which elucidates the in vivo roles of these enzymes in setting up DNA methylation patterns. We show that severely hypomethylated embryos overexpress a panel of genes, transposons, and illegitimate transcripts initiating from cryptic promoters, revealing the multiple roles of DNA methylation for the maintenance of transcriptional integrity in development.

## Results

**Methylome profiling of Dnmt mutant embryos.** To assess the contribution of DNMTs to DNA methylation in vivo, we generated base-resolution methylomes in Dnmt mutant embryos. Using a Dnmt1-2lox allele[30], we created Dnmt1 mutant embryos lacking the exons 4 and 5, which creates an out-of-frame splice and a functional null allele. As shown previously[19], the Dnmt1$^{-/-}$ embryos showed growth retardation at embryonic day (E) 8.5 (Fig. 1a). For DNMT3A and DNMT3B, we previously showed that methylation is only partially reduced in single knockouts, suggesting redundancy[7]. To address this, we generated Dnmt3a/b DKO embryos. Confirming previous observations[4], DKO embryos resembled Dnmt1-null embryos and showed growth retardation at E8.5 (Fig. 1a). We performed MspI-based reduced representation bisulfite sequencing (RRBS) in three Dnmt1$^{-/-}$ and WT littermate controls at E8.5, as well as three DKO embryos and controls from the same litters (Supplementary Table 1). The RRBS data were highly reproducible between replicates (Supplementary Fig. 1a–c). To have a complete view, we also performed whole genome bisulfite sequencing (WGBS) in two independent WT, Dnmt1$^{-/-}$ and DKO E8.5 embryos (Supplementary Table 2). The average sequencing depth after deduplication was 12× and close to 90% of the CpGs were sequenced at least 5× in each dataset (Supplementary Fig. 1d and Supplementary Table 2). The WGBS data were reproducible between independent embryos (Supplementary Fig. 1e, f), demonstrating the reliability of the datasets.

**DNMT1 is required to sustain DNA methylation genome-wide.** We found a strong reduction of genomic methylation in Dnmt1$^{-/-}$ embryos (Fig. 1b). The mean CG methylation level measured by RRBS in non-CGI sequences dropped from 69.8% in WT to 16.7% in Dnmt1$^{-/-}$ embryos, whereas CG methylation of CGIs dropped from 2.4% in WT to 0.7% in Dnmt1$^{-/-}$ embryos (Fig. 1c). Confirming the RRBS results, the global CG methylation levels measured by WGBS in WT and Dnmt1$^{-/-}$ embryos were 81.6% and 20.3%, respectively (Fig. 1d, e). The loss of methylation upon inactivation of Dnmt1 is truly global and occurs across all genomic sequences including exons, introns, intergenic regions and TEs (Fig. 1f, g). All sequences have low to intermediate methylation and no sequences retain high methylation in Dnmt1$^{-/-}$ embryos (Fig. 1h, i), indicating that DNMT1 is universally required to sustain DNA methylation at all genomic sequences in embryonic development.

**De novo DNA methylation is abolished in Dnmt3a$^{-/-}$ Dnmt3b$^{-/-}$ embryos.** Next, we analyzed the methylome of Dnmt3a/b DKO embryos. In contrast to the single knockouts[7], the double inactivation of Dnmt3a/b lead to a strong reduction of DNA methylation, demonstrating redundancy between DNMT3A and DNMT3B genome-wide (Fig. 1b). The mean CG methylation level measured by RRBS in non-CGI sequences dropped from 69.8% in WT to 63.0% in Dnmt3a$^{-/-}$, 49.6% in Dnmt3b$^{-/-}$, and 15.0% in DKO embryos (Fig. 1c). Accordingly, the CG methylation level measured by WGBS dropped from 81.6% in WT to 18.2% in DKO embryos (Fig. 1d, e). The loss of methylation in DKO embryos affects all genome compartments (Fig. 1f). However, in contrast to Dnmt1$^{-/-}$ embryos, DKO embryos do not display a uniform loss of methylation (Fig. 1g) but contain a high proportion of sequences fully demethylated or highly methylated (Fig. 1h, i).

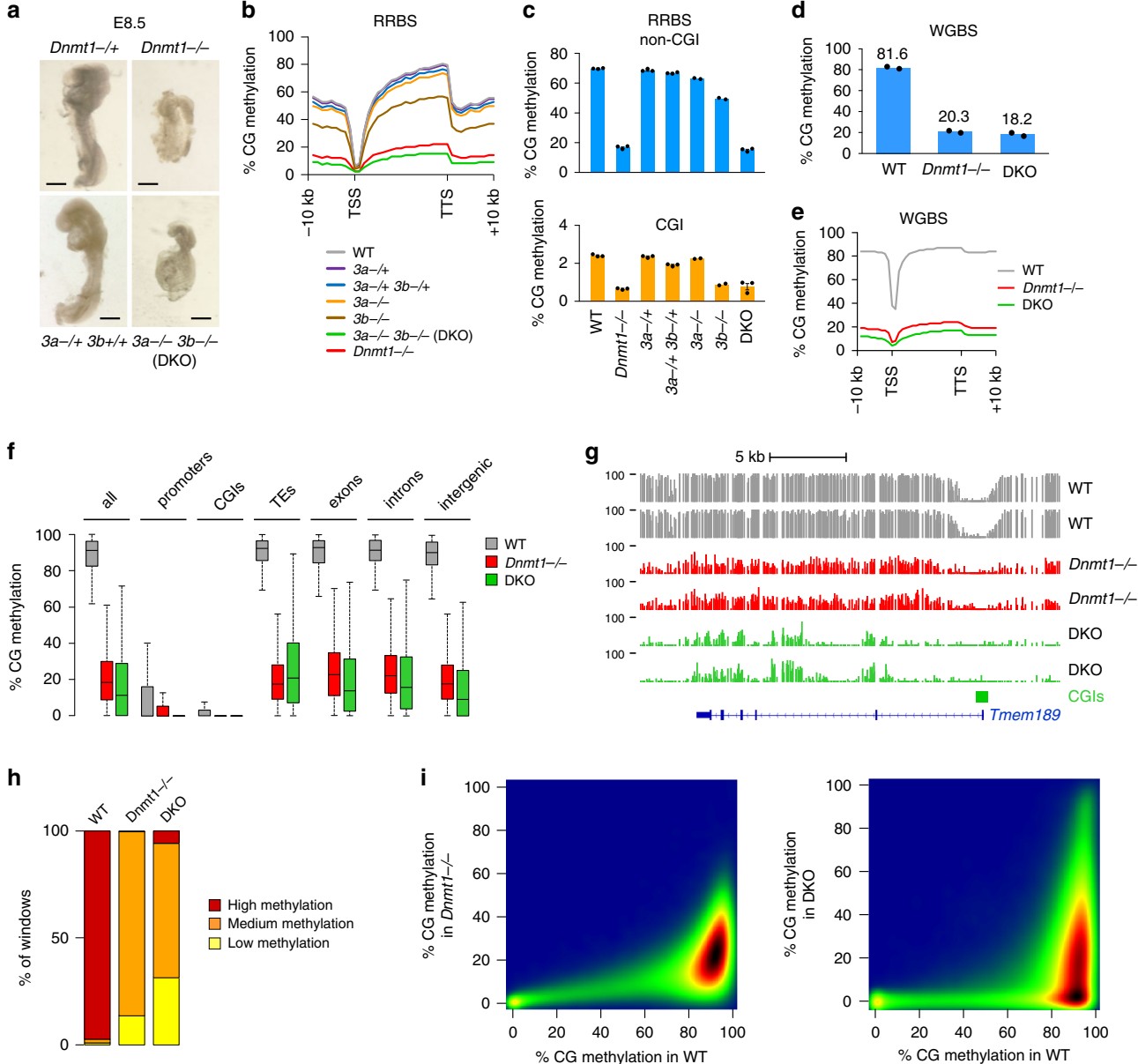

**Fig. 1 Methylome profiling in _Dnmt_ mutant embryos. a** Images of _Dnmt1−/−_ and DKO embryos dissected at E8.5 compared with littermate heterozygous controls (which are identical to WT). Black bars: 300 μm. **b** Average distribution of CG methylation over RefSeq genes and 10 kb flanking sequences calculated from the RRBS data in E8.5 embryos. Our previous data from single _Dnmt3a−/−_ and _Dnmt3b−/−_ embryos[7] was included for comparison. TSS transcription start site, TTS transcription termination site. **c** Average methylation of CGs located outside of CpG islands (non-CGI, top) or within CpG islands (CGI, bottom) measured by RRBS in _Dnmt_ mutant and control embryos (mean ± SEM, _n_ = 3 independent embryos for all genotypes except _n_ = 2 for _Dnmt3a−/−_ and _Dnmt3b−/−_). **d** Bar plot representing the average CG methylation level measured by WGBS in WT, _Dnmt1−/−_ and DKO E8.5 embryos (mean of _n_ = 2 independent embryos). **e** Metaplots of CG methylation levels in RefSeq genes and 10 kb flanking sequences calculated from the WGBS data in WT, _Dnmt1−/−_ and DKO E8.5 embryos. **f** Boxplots of CG methylation levels measured by WGBS in different genomic features in WT, _Dnmt1−/−_ and DKO E8.5 embryos. In the boxplots the line indicates the median, the box limits indicate the upper and lower quartiles and the whiskers extend to 1.5 IQR from the quartiles. **g** Example of genome browser view of WGBS methylation profiles in two independent replicates of WT, _Dnmt1−/−_ and DKO E8.5 embryos. Each track shows the percent methylation of individual CpGs between 0 and 100%. CpG islands (CGIs) depicted by green rectangles and RefSeq gene annotations are shown below the tracks. **h** Stacked bar graph representing the proportions of genomic windows (1 kb) with high (>50%), medium (10–50%), and low (<10%) CG methylation in WT, _Dnmt1−/−_ and DKO embryos. **i** Density scatter plots comparing WGBS methylation scores in 500 bp windows between WT, _Dnmt1−/−_ and DKO E8.5 embryos. The density of points increases from blue to dark red. In **f**, **h**, **i**, values are the average of _n_ = 2 independent embryos. Source data are provided as a Source data file.

To determine the origin of high methylation in DKO embryos, we compared the WGBS methylation patterns of DKO embryos to those of preimplantation inner cell mass (ICM)[31]. Strikingly, visual inspection of the methylation patterns in ICM and DKO embryos revealed that they are highly similar (Fig. 2a,

Supplementary Fig. 2a). To confirm this, we performed a pairwise comparison of WGBS methylation scores and revealed a strong positive correlation between methylation in ICM and DKO embryos (Fig. 2b, _r_ = 0.80). We found that the most highly methylated sequences in DKO embryos are enriched for TEs

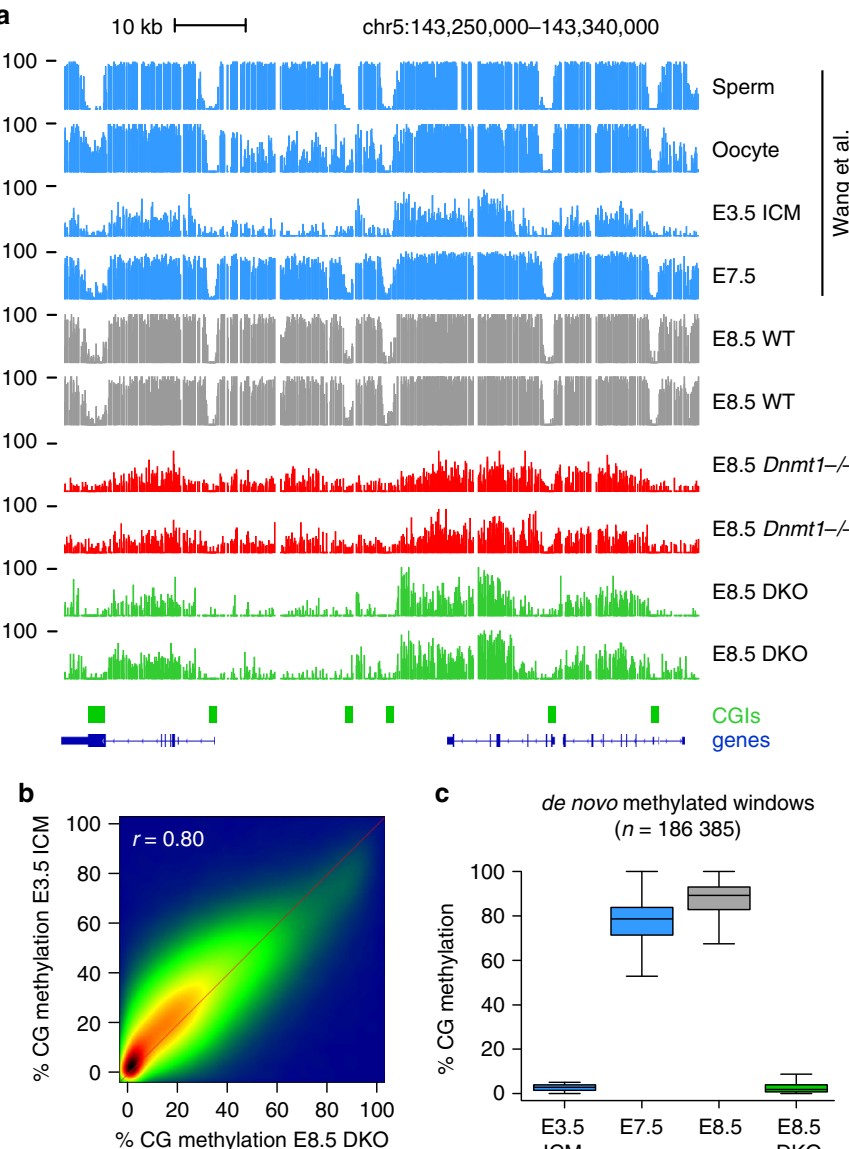

**Fig. 2 Absence of de novo methylation in DKO embryos. a** Genome browser tracks of WGBS methylation profiles in gametes, early embryos[31], and E8.5 *Dnmt* mutant embryos, highlighting the strong similarity of methylation patterns between the ICM (Inner Cell Mass) and DKO embryos. CpG islands (CGIs) depicted by green rectangles and RefSeq gene annotations are shown below the tracks. **b** Density scatter plot of WGBS methylation scores in 1 kb windows in E8.5 DKO embryos compared with E3.5 ICM. The density of points increases from blue to dark red. **c** Boxplots of the CG methylation levels in ICM, postimplantation embryos, and E8.5 DKO embryos for all genomic windows de novo methylated at implantation, illustrating that de novo methylation is abolished in DKO embryos. In the boxplots the line indicates the median, the box limits indicate the upper and lower quartiles and the whiskers extend to 1.5 IQR from the quartiles.

(Supplementary Fig. 2b). Amongst the most methylated TE families are IAP, RLTR6, and MMERVK10C elements, which carry high methylation in DKO embryos at levels identical to blastocysts (Supplementary Fig. 2c, d). This strongly suggests that in absence of DNMT3A/B, methylation in E8.5 DKO embryos arises from the maintenance of preexisting DNA methylation from the blastocysts. Finally, to determine if de novo methylation happens in DKO embryos, we selected all the sequences that are hypomethylated in blastocysts and gain methylation in E8.5 embryos, and found that de novo methylation at these sequences is completely abolished in DKO embryos (Fig. 2c). Altogether this indicates that DNMT3A/B are responsible for the bulk de novo DNA methylation between the blastocyst and postimplantation stages and that DNMT1 has a negligible capacity for de novo DNA methylation during embryonic development.

**Role of DNMT3A/B in methylation maintenance**. Given the proposed function of DNMT3A/B in maintenance DNA methylation in ES cells, we investigated the role of DNMT3A/B in maintenance DNA methylation in vivo. First, we quantified methylation of 20 known germline DMRs (gDMRs) of imprinted loci, which arise from the postfertilization maintenance of allelic methylation established in the parental gametes (Supplementary Data 1). The methylation of all gDMRs is preserved in DKO embryos (Fig. 3a, b), indicating that DNMT3A/B have no significant contribution to the maintenance of gametic-derived methylation imprints in embryos. This agrees with two previous studies showing that DNMT3A/B are dispensable for maintenance methylation of the *Igf2r*, *H19*, and *Dlk1/Gtl2* gDMRs in vivo[6,18]. In contrast, all gDMRs are demethylated in *Dnmt1*[-/-] embryos, confirming that DNMT1 is the main enzyme propagating

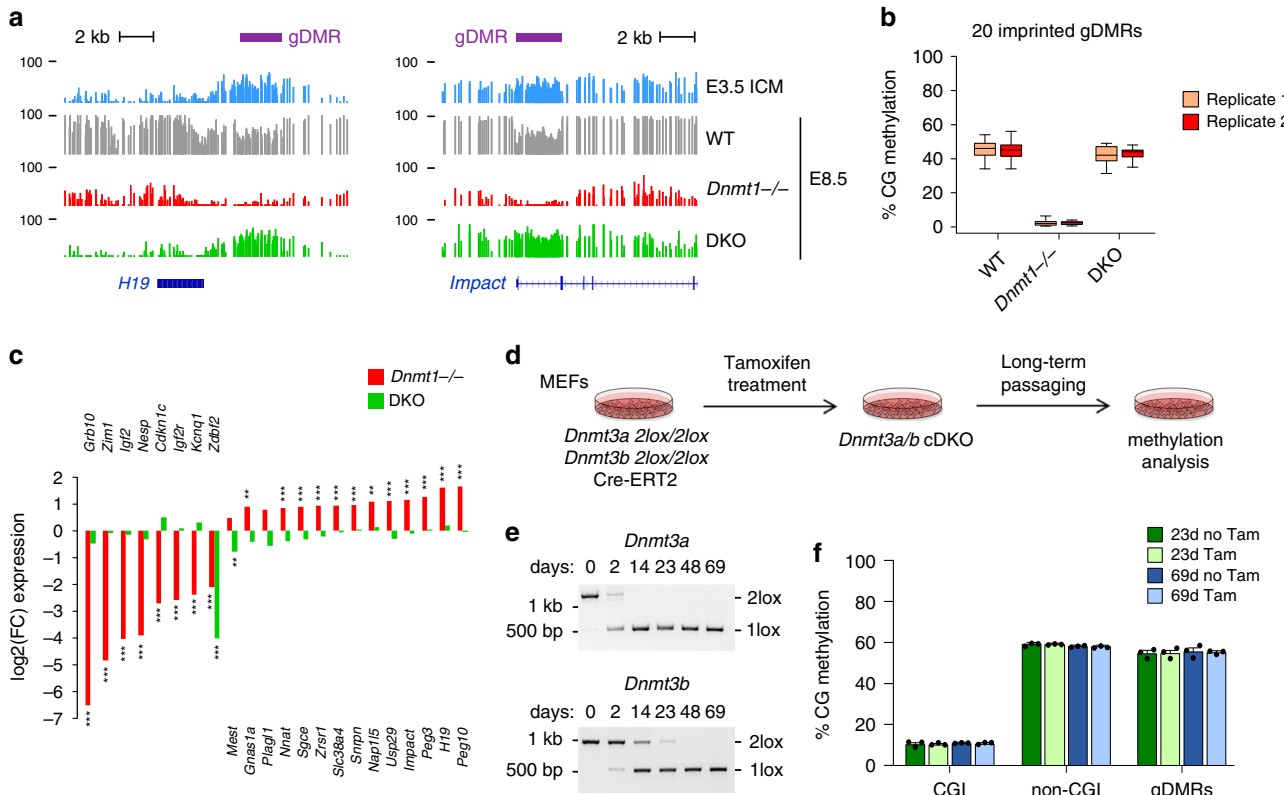

**Fig. 3 Role of DNMTs in methylation imprints and maintenance methylation. a** Genome browser tracks of WGBS methylation profiles at the *H19* and *Impact* imprinted loci in early embryos and *Dnmt* mutant embryos. One WGBS replicate is shown. The positions of the imprinted germline DMRs (gDMRs) are depicted by purple rectangles. **b** Boxplots of the CG methylation levels of $n = 20$ imprinted germline DMRs measured by WGBS in WT, *Dnmt1*$^{-/-}$ and DKO embryos. In the boxplots the line indicates the median, the box limits indicate the upper and lower quartiles and the whiskers extend to 1.5 IQR from the quartiles. **c** Fold change of expression of imprinted genes in *Dnmt1*$^{-/-}$ (red bars) and DKO (green bars) embryos ($n = 3$ embryos for *Dnmt1*$^{-/-}$; $n = 6$ embryos for DKO). **$p < 0.01$; ***$p < 0.001$ (adjusted $p$ value calculated by DESeq2 using a Wald test corrected for multiple testing). **d** Experimental outline for investigating the maintenance function of DNMT3A/B in MEFs. *Dnmt3a*$^{2lox/2lox}$ *Dnmt3b*$^{2lox/2lox}$ immortalized MEFs expressing Cre-ERT2 were treated with Tamoxifen to generate conditional double knockout (cDKO) cells, and methylation was quantified by RRBS after long-term passaging. **e** Evaluation by PCR genotyping of the Cre-mediated recombination of the *Dnmt3a* and *Dnmt3b*-2lox alleles in cDKO fibroblasts. The number of days of Tamoxifen treatment is indicated above the gels. **f** CG methylation levels quantified by RRBS in cDKO fibroblasts. The graph shows the methylation levels in CpG islands (CGI), non-CGI regions, and imprinted gDMRs in cells treated with Tamoxifen (Tam) or not treated with Tamoxifen (no Tam) after 23 and 69 days of culture (mean ± SEM, $n = 3$ biological replicates), revealing no global hypomethylation in cDKO MEFs. Source data are provided as a Source data file.

methylation imprints in embryos (Fig. 3a, b). Moreover, extending previous findings[20,21], RNA-seq in *Dnmt* mutant embryos (see below) revealed marked changes in expression of imprinted genes in *Dnmt1*$^{-/-}$ but not DKO embryos, including, as expected, both downregulation (*Grb10*, *Igf2*, and *Kcnq1*) and ~2-fold upregulation (*Snrpn*, *Peg3*, and *H19*) (Fig. 3c). The exception is *Zdbf2* that shows reduced expression in DKO embryos (Fig. 3c), validating the model that embryonic de novo methylation is required for an unusual switch from a maternal to paternal DMR and activate paternal-specific transcription of *Zdbf2*[32]. In summary, we show that DNMT1 alone mediates maintenance of methylation imprints and provide in vivo validation for the role of DNA methylation at many imprinted loci.

To investigate a possible maintenance function of DNMT3A/B at other genomic loci, it is necessary to perform conditional inactivation after de novo methylation has been completed. To this aim, we derived and immortalized *Dnmt3a*$^{2lox/2lox}$ *Dnmt3b*$^{2lox/2lox}$ MEFs and generated *Dnmt3a/b* conditional double knockouts (cDKO) with a tamoxifen-inducible CRE recombinase (Cre-ERT2) (Fig. 3d). Tamoxifen treatment led to efficient recombination of the *Dnmt3a* and *Dnmt3b* alleles (Fig. 3e). Furthermore, RT-qPCR confirmed that the expression of the floxed exons of *Dnmt3a/b* became undetectable following

tamoxifen treatment in cDKO MEFs while *Dnmt1* expression is unchanged (Supplementary Fig. 3a, b). The strong reduction of DNMT3A protein in cDKO cells was validated by western blot (Supplementary Fig. 3c), whereas we could not detect DNMT3B by western blot even in untreated MEFs consistently with the known lack of DNMT3B expression in differentiated cells. The cells were cultivated for up to 69 days to allow multiple cell divisions. RRBS performed at 23 days and 69 days of culture (Supplementary Table 3) revealed no evidence of decreased methylation genome-wide (Fig. 3f, Supplementary Fig. 3d). After 69 days of culture, we only identified 11 hypomethylated DMRs in *Dnmt3a/b* cDKO fibroblasts that most likely reflect de novo methylation events that happened during the course of cell culture (Supplementary Fig. 3e, f). As a control, we also derived and immortalized *Dnmt1*$^{2lox/2lox}$ MEFs to generate *Dnmt1* conditional knockout MEFs with Cre-ERT2 (Supplementary Fig. 3g–i). Conditional inactivation of *Dnmt1* led to an immediate and global hypomethylation of genomic DNA as measured by RBBS after 5 and 7 days of tamoxifen treatment (Supplementary Fig. 3j, k) associated with a block of cellular division, confirming that DNMT1 is the sole maintenance enzyme. Taken together, our data indicate no major role of DNMT3A/B in maintenance methylation in embryos and differentiated cells.

**DNA methylation suppresses the germline program in embryos**. Next, we used RNA-seq to investigate the consequences of the absence of DNA methylation on gene expression in embryos. RNA-seq was performed on three $Dnmt1^{-/-}$ and WT littermate embryos, as well as six DKO embryos and six WT and $Dnmt3a^{-/+}$ littermate controls (Supplementary Table 4). RNA-seq confirmed the knockout of critical exons in the $Dnmt$ genes (Supplementary Fig. 4a, b). This analysis identified 414 upregulated and 68 downregulated genes in $Dnmt1^{-/-}$ embryos, and 564 upregulated and 47 downregulated genes in DKO embryos (fold change > 3, DESeq2 adjusted $p$ value < 0.001) (Supplementary Fig. 4c, Supplementary Data 2). Principal component analysis showed high similarity between $Dnmt1^{-/-}$ and DKO samples, which cluster separately from the controls (Fig. 4a). Indeed, there is a good correlation between the expression changes in $Dnmt1^{-/-}$ and DKO embryos (Supplementary Fig. 4d), and the genes misregulated in $Dnmt1^{-/-}$ and DKO embryos strongly overlap (Fig. 4b). In contrast, these genes only weakly overlap with the genes upregulated in TKO mESCs (Supplementary Fig. 4e, f), mostly because many show a strong basal expression in WT ESCs (Supplementary Fig. 4g).

We then focused on the genes upregulated in DKO embryos and classified them in three groups: group 1 includes genes with low CpG promoters (LCP), group 2 includes genes with unmethylated CpG-rich promoters (intermediate or high CpG promoters, ICP or HCP), and group 3 includes genes with methylated CpG-rich promoters (Fig. 4c). The genes of group 2 show only weak derepression compared with the other groups and could partly reflect indirect effects (Supplementary Fig. 4h). Whereas no significant ontology terms are associated with the groups 1 and 2, the group 3 is strongly enriched for ontology terms related to germline functions (Fig. 4c). In total, 137 germline genes acquire dense promoter CpG methylation in WT embryos and are derepressed in DKO embryos (Supplementary Data 2), with some (e.g., $Tuba3b$, $Sohlh2$, $Tex13$, $Rpl10l$, $Dazl$, $Asz1$, and $Hormad1$) reaching up to ~1000-fold upregulation (Fig. 4d). This includes numerous germ-cell specific piRNA pathway factors ($Gtsf1$, $Tex19$, $Topaz1$, $Rnf17$, $Piwil2$, $Mov10l1$, $Asz1$, $Ddx4$, $Mael$, $Fkbp6$, and $Gpat2$). Interestingly, we previously found some germline genes upregulated in $Dnmt3b^{-/-}$ embryos[7,22]. As expected, all these genes are also upregulated in DKO embryos, nevertheless many additional germline genes are derepressed in DKO embryos (Supplementary Fig. 4i). Moreover, the degree of reactivation of germline genes is much higher in DKO embryos, which correlates with the degree of methylation loss (Supplementary Fig. 4j). This supports a direct relationship between CpG-island promoter methylation and repression of a large panel of germline genes.

To firmly demonstrate that local CpG-island methylation mediates repression of germline genes, we performed dCas9-based targeted demethylation with the TET1 catalytic domain (TET1CD) in MEFs using gRNAs targeting the $Dazl$ and $Asz1$ promoters (Supplementary Fig. 5a, b). We first compared the efficiency of dCas9-TET1CD fusion and the dCas9-SunTag-TET1CD system[33] and found that only dCas9-SunTag-TET1CD achieved robust demethylation of $Dazl$ and $Asz1$ (Supplementary Fig. 5c–e). Targeted demethylation with dCas9-SunTag-TET1CD induced strong derepression of $Dazl$ and $Asz1$ (Fig. 4e), demonstrating that dense promoter methylation of germline genes plays a causal role in the maintenance of their repressed state. In summary, we reveal an extensive role of DNA methylation in keeping CpG-rich promoters of the germline program silent in embryos.

Given the derepression of germline genes in hypomethylated embryos, we wondered what their expression is in hypomethylated blastocysts. We analyzed RNA-seq from E3.5 ICM[34] and found that, while approximately 1/3 of the most derepressed germline genes in DKO embryos have abundant mRNAs in ICM, the majority has low or undetectable expression in ICM (Supplementary Fig. 6a). For example, the $Dazl$ and $Slc25a31$ genes show weak expression in E3.5 ICM despite similar hypomethylation than in E8.5 DKO embryos (Supplementary Fig. 6b). This suggests that either activators of germline genes are absent in blastocysts or that transient repression mechanisms compensate for erased DNA methylation in preimplantation stages before a switch to DNA methylation-dependent repression in postimplantation embryos.

**DNA methylation limits early expression of lineage-committed genes**. We then investigated what other genes are in the overexpressed groups (Fig. 4c). Besides expected targets such as $Rhox$ genes[6], we found that several somatic lineage-committed genes harboring a CpG-dense promoter acquire promoter DNA methylation in WT embryos and are overexpressed in DKO and $Dnmt1^{-/-}$ embryos. These genes belong in majority to the group 3 and include genes expressed in hematopoietic cells ($Bin2$, $Arhgap30$, $Ly86$, $Pf4$, and $Nckap1l$), brain ($A330102I10Rik$), eye ($Rbp3$), or digestive tissues ($Iyd$, $Gstp2$) (Fig. 4f, Supplementary Data 2). Their overexpression was validated by RT-qPCR in DKO embryos (Fig. 4g). This suggests that DNA methylation of CpG-dense promoters contributes to prevent ectopic expression of some lineage-committed genes. One prediction of this model is that promoter DNA methylation of these genes should be low in the tissues where they are expressed. To test this prediction, we explored public WGBS data from mouse tissues[35–37] and confirmed that the promoters of these genes are specifically hypomethylated in the tissues where they are expressed (Fig. 4h, Supplementary Fig. 7). Taken together, this suggests a role of DNA methylation in suppressing precocious expression of lineage-committed genes in embryos.

**De novo DNA methyltransferases are required to repress 2C-specific genes**. We noticed that many genes specifically expressed in 2-cell embryos and 2C-like ES cells (2C-genes) are strongly derepressed (up to ~1000-fold) in DKO embryos, such as $Zscan4$ genes, $Tmem92$, $Tcstv1/3$, and $Eif1a$-like genes ($Gm5662$, $Gm2022$, BB287469, $Gm2016$, $Gm21319$, $Gm8300$, and $Gm5039$) and $Usp17$-like genes (Fig. 4i), whose differential expression was validated by RT-qPCR (Fig. 4j). To confirm this finding, we compared the list of genes upregulated in DKO embryos with genes upregulated in 2C-like ES cells[38] and found a significant overlap (Supplementary Fig. 8a). These 2C-genes are frequently organized in clusters (Supplementary Fig. 8b). Intriguingly, the extent of upregulation of 2C-genes is variable between DKO embryos (Fig. 4i). Moreover, most of these 2C-genes contain CpG-poor promoters and are much less overexpressed in $Dnmt1^{-/-}$ embryos compared with DKO embryos despite similar hypomethylation (Fig. 4i, Supplementary Fig. 8c), suggesting a possible role of de novo methyltransferases independent of DNA methylation. Previous studies have established that another feature of 2C embryos is the expression of MERVL retrotransposons and that many 2C-genes initiate from promoters in ERVL LTRs[39–41]. Accordingly, we found a threefold activation of MERVL-int transposons in DKO embryos but not in $Dnmt1^{-/-}$ embryos (Supplementary Fig. 8d). In addition, several 2C-specific genes derepressed in DKO embryos (i.e., $Zfp352$, $Tcstv1$, $Tcstv3$, $B020031M17Rik$, AF067061, $Gm20767$, $Ubtfl1$, $Gm2022$, and AA792892) initiate from MERVL LTRs (annotated as MT2_Mm) or other ERVL transposons (MT2B-C, ORR1B) (Supplementary Fig. 8e). Thus de novo methyltransferases are required for the

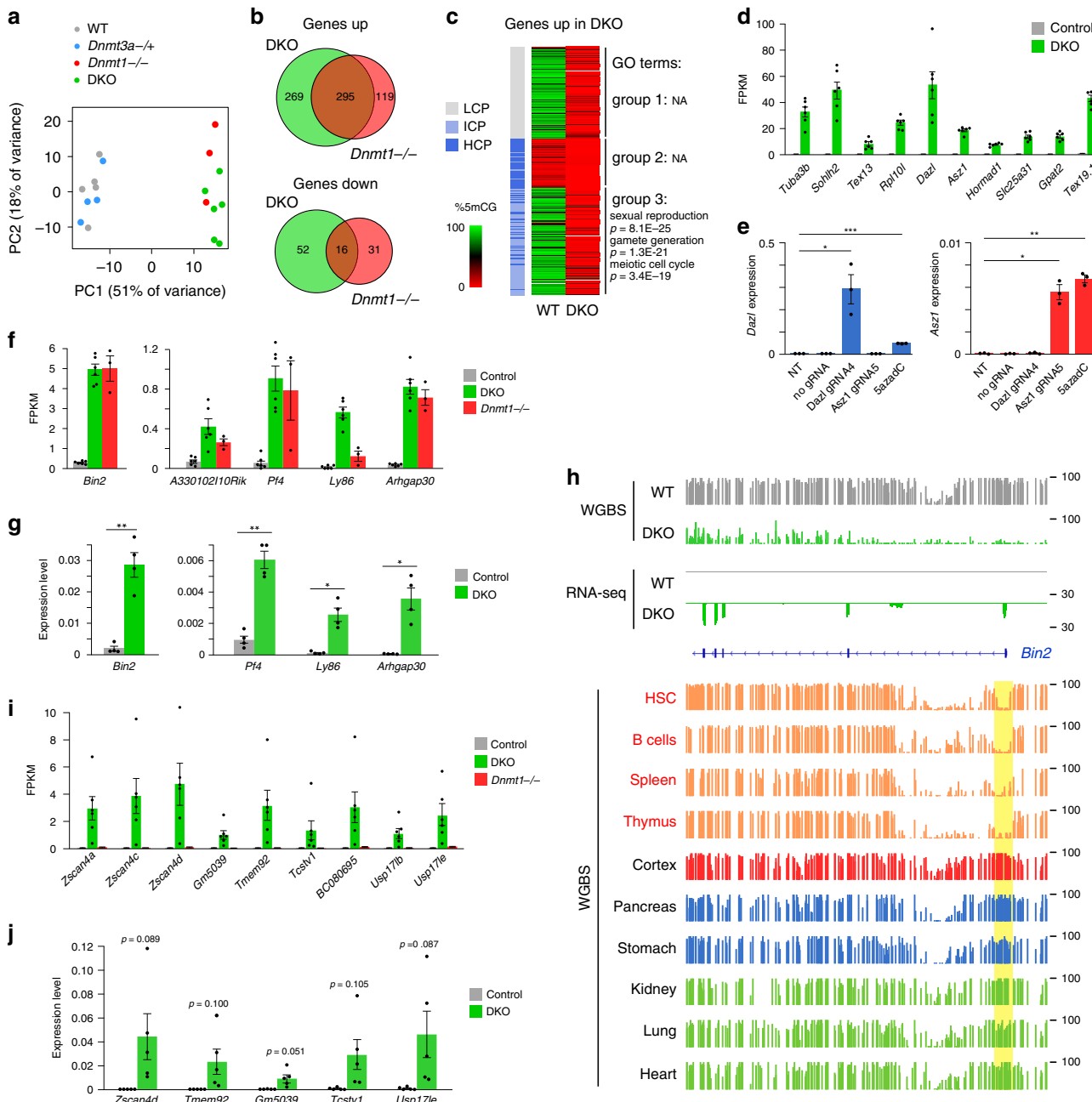

**Fig. 4 Transcriptome analysis of *Dnmt* mutant embryos. a** Principal component analysis of RNA-seq data. **b** Venn diagram comparing the lists of upregulated and downregulated genes in *Dnmt1*$^{-/-}$ and DKO embryos. **c** Heatmap of the three groups of genes upregulated in DKO embryos classified by their promoter class (LCP low CpG promoter, ICP intermediate CpG promoter, HCP high CpG promoter) and promoter methylation in WT embryos (measured in −1000 to +500 bp from the TSS). Group 1: LCP; Group 2: ICP or HCP and promoter methylation < 30%; Group 3: ICP or HCP and promoter methylation ≥ 30%. Gene ontology (GO) terms enriched in each group are shown on the right. **d** Expression levels (FPKM) of germline genes in DKO and control embryos (mean ± SEM, $n = 6$ embryos). **e** RT-qPCR expression levels of *Dazl* and *Asz1* in non-transfected MEFs (NT) and MEFs expressing dCas9-Suntag-TET1 with no gRNA or a gRNA targeting the *Dazl* or *Asz1* promoter (mean ± SEM, $n = 3$ independent experiments). Cells treated with 0.5 μM 5-Aza-2′-deoxycytidine (5azadC) for 72 h were used as control. *$p < 0.05$; **$p < 0.01$; ***$p < 0.001$ (two-tailed unpaired *t* test). **f** Expression levels (FPKM) of lineage-committed genes in DKO, *Dnmt1*$^{-/-}$ and control embryos (mean ± SEM, $n = 6$ embryos for control and DKO, $n = 3$ for *Dnmt1*$^{-/-}$). **g** RT-qPCR analysis of the expression of five lineage-committed genes in control and DKO embryos (mean ± SEM, $n = 4$ embryos). *$p < 0.05$; **$p < 0.01$; ***$p < 0.001$ (two-tailed unpaired *t* test). **h** Genome browser tracks of RNA-seq and WGBS at the hematopoietic-specific *Bin2* gene in WT and DKO embryos and adult tissues[35-37]. For embryos, one replicate of RNA-seq and WGBS is shown. The *Bin2* promoter (highlighted in yellow) is specifically hypomethylated in hematopoietic tissues (written in red). **i** Expression levels (FPKM) of 2C-specific genes in DKO, *Dnmt1*$^{-/-}$ and control embryos (mean ± SEM, $n = 6$ embryos for control and DKO, $n = 3$ for *Dnmt1*$^{-/-}$). **j** RT-qPCR analysis of the expression of 2C-specific genes in control and DKO embryos (mean ± SEM, $n = 5$ embryos). The *p* values are indicated (two-tailed unpaired *t* test). Source data are provided as a Source data file.

extinction of the ERVL-driven and 2C-specific transcriptional network in postimplantation embryos.

**DNA methylation represses a high number of TEs and chimeric transcripts**. The contribution of DNA methylation to the regulation of TEs in the embryo has not been studied comprehensively, which prompted us to analyze the expression of TEs in *Dnmt* mutant embryos. TE expression was quantified either by counting reads in RepeatMasker annotations or by mapping reads on Repbase sequences (see "Methods") (Supplementary Data 3). IAPs showed a dramatic reactivation (50–100 fold) in *Dnmt1*$^{-/-}$ embryos (Fig. 5a), confirming previous data by northern blot and in situ hybridization[17]. In addition to IAPs, several other retrotransposon families of the LINE-1, ERV1, and ERVK families were significantly upregulated in *Dnmt1*$^{-/-}$ embryos (Fig. 5a, Supplementary Fig. 9a). The same set of transposons was upregulated in DKO embryos but with a lower magnitude, which correlates with higher residual methylation of TEs in DKO embryos (Fig. 5a). The RepBase method yielded similar results (Supplementary Fig. 9b) and revealed that among the LINE-1 elements, only the most recent subfamilies are upregulated (Supplementary Fig. 9c). Having shown that DNA methylation is required to repress TEs at the family level, we analyzed the expression of individual copies of TEs by using uniquely mapped reads. It should be mentioned that this method underestimates the counts of upregulated TE copies because very young TEs cannot be uniquely mapped. This analysis identified 4593 activated TE copies (fold change > 3, DESeq2 adjusted *p* value < 0.001) in *Dnmt1*$^{-/-}$ embryos (Supplementary Fig. 9d–g, Supplementary Data 3). For IAPs, the most active retroelements in the mouse, some families showed a massive reactivation of up to 30% annotated copies (Fig. 5b), in particular IAPEz-int that represent more than half (2484 out of 4593) of all upregulated TEs in *Dnmt1*$^{-/-}$ embryos (Supplementary Fig. 9f). Other ERVs like ERVB4_1B-I_MM-int, MMERGLN-int, MMEtn-int, and MMERVK10C-int show activation of a limited set of copies representing no more than 4% of all annotated copies (Fig. 5b). Interestingly, these activated copies have a higher size and presumably correspond to full length, potentially active copies (Fig. 5c).

Next, we investigated the impact of the derepression of ERVs on the expression of neighboring genes. We identified 715 genes located close to activated ERVs (<20 kb from the TSS) and found that they were significantly more upregulated than control genes in *Dnmt1*$^{-/-}$ embryos (Fig. 5d, e), indicating that derepressed ERVs alter the expression of proximal genes. Out of the 414 upregulated genes in *Dnmt1*$^{-/-}$ embryos, 10% (*n* = 42 genes) are upregulated in association with derepression of an intragenic or proximal ERV (Supplementary Data 2). In some cases, intergenic ERVs initiate long RNAs that extend into adjacent genes and produce chimeric transcripts by splicing to an internal exon, as exemplified by the *Cyp2b23*, *Serpinb1c*, and *Olfr316* genes (Fig. 5f, Supplementary Fig. 10a, b). We also observed intragenic initiation from intronic IAPEz and its flanking LTR IAPLTR1_Mm inserted in the antisense orientation to the host gene, as exemplified by the *Capn11*, *Trpm2*, and *Apoh* genes (Fig. 5g, Supplementary Fig. 10c, d). To determine if antisense transcription from IAPLTR1_Mm elements is frequent, we counted the RNA signal from all IAPLTR1_Mm elements and found that, while there is a strong activation in the sense orientation, there is also a noticeable increase in antisense transcription initiation from these elements in *Dnmt1*$^{-/-}$ embryos (Supplementary Fig. 10e, f). Altogether this shows that derepressed TEs alter the expression of nearby genes, similarly to previous observations made in *Setdb1* KO ES cells[23] and *Dnmt3L* mutant

spermatocytes[42]. Interestingly, in contrast to *Dnmt3L* KO spermatocytes[42], we did not see activation of nearby genes by L1 retrotransposons, suggesting that the impact of hypomethylated TEs on the integrity of the transcriptome is different between germ cells and somatic cells.

In summary, we conclude that DNMT1 is the main enzyme involved in TE protection and that DNA methylation is required to repress intact, potentially active copies of many retrotransposon families and prevent them from disturbing expression of nearby genes.

**DNA methylation suppresses cryptic initiation sites in gene bodies**. Upon further exploring the transcriptional changes in mutant embryos, we noticed that several genes upregulated in DKO embryos initiate from intragenic sequences not associated with TEs. For example, the *Mgl2*, *Mlana*, *C8b*, and *Plekhd1* genes produce truncated mRNAs initiating in the gene body (Fig. 6a, Supplementary Fig. 11a–c). To measure if cryptic intragenic transcription initiation is a general phenomenon, we calculated the ratio of expression of downstream exons versus the first exon for all genes. This revealed no significant increase in DKO compared with WT embryos (Fig. 6b), suggesting that cryptic internal initiation is limited to a subset of genes. Using a bioinformatic pipeline (see "Methods"), we identified 46 genes in DKO and 25 genes in *Dnmt1*$^{-/-}$ embryos upregulated from intragenic sequences not annotated as transposons or alternative promoters (Supplementary Data 2). Consistent with a primary role of DNA methylation, the genes identified in DKO and *Dnmt1*$^{-/-}$ embryos largely overlapped (Supplementary Fig. 11d). The sites of cryptic intragenic initiation in these genes tend to be CpG-rich and, in contrast to canonical promoter sequences, are strongly methylated in WT embryos (Fig. 6c).

To explore the mechanisms of intragenic initiation, we focused on the *Mgl2* gene. Interestingly, the activation of the cryptic *Mgl2* promoter is recapitulated in TKO ES cells (Fig. 6d), making these cells a good model to investigate *Mgl2* regulation. Analysis of previous datasets generated in TKO ES cells[24] revealed that the absence of DNA methylation is associated with the appearance of a new DNase-I hypersensitive site and binding of the methylation-sensitive transcription factor NRF1 at the site of intragenic initiation in the intron 6 (Fig. 6d). The sequence of this intron contains three repetitions of the NRF1 binding motif GAGCATGCGC (Supplementary Fig. 11e). This suggests that internal binding of NRF1 in absence of DNA methylation creates an intragenic initiation site in the *Mgl2* gene. To validate this hypothesis, we monitored *Mgl2* expression in TKO ES cells knocked down for NRF1 and found that *Mgl2* internal initiation is abolished in these cells (Fig. 6e). Taken together, this reveals that DNA methylation is critical to prevent methylation-sensitive transcription factors from creating cryptic intragenic initiation sites in embryos.

**Discussion**
In this paper, we interrogated the contribution of DNMTs to the establishment of genome-wide DNA methylation patterns in mouse embryos. Our results support a strict division of function between DNMT1 and DNMT3A/B in vivo. DNMT1 alone mediates the faithful maintenance of DNA methylation in developing embryos with no contribution of DNMT3A/B, as supported by several lines of evidence: (1) DNMT1 alone is sufficient to maintain preexisting patterns of DNA methylation from the blastocysts to the E8.5 stage in DKO embryos; (2) All gametic methylation imprints are faithfully maintained in DKO embryos; (3) Global patterns of DNA methylation are unaffected upon conditional inactivation of *Dnmt3a/b* over multiple cell

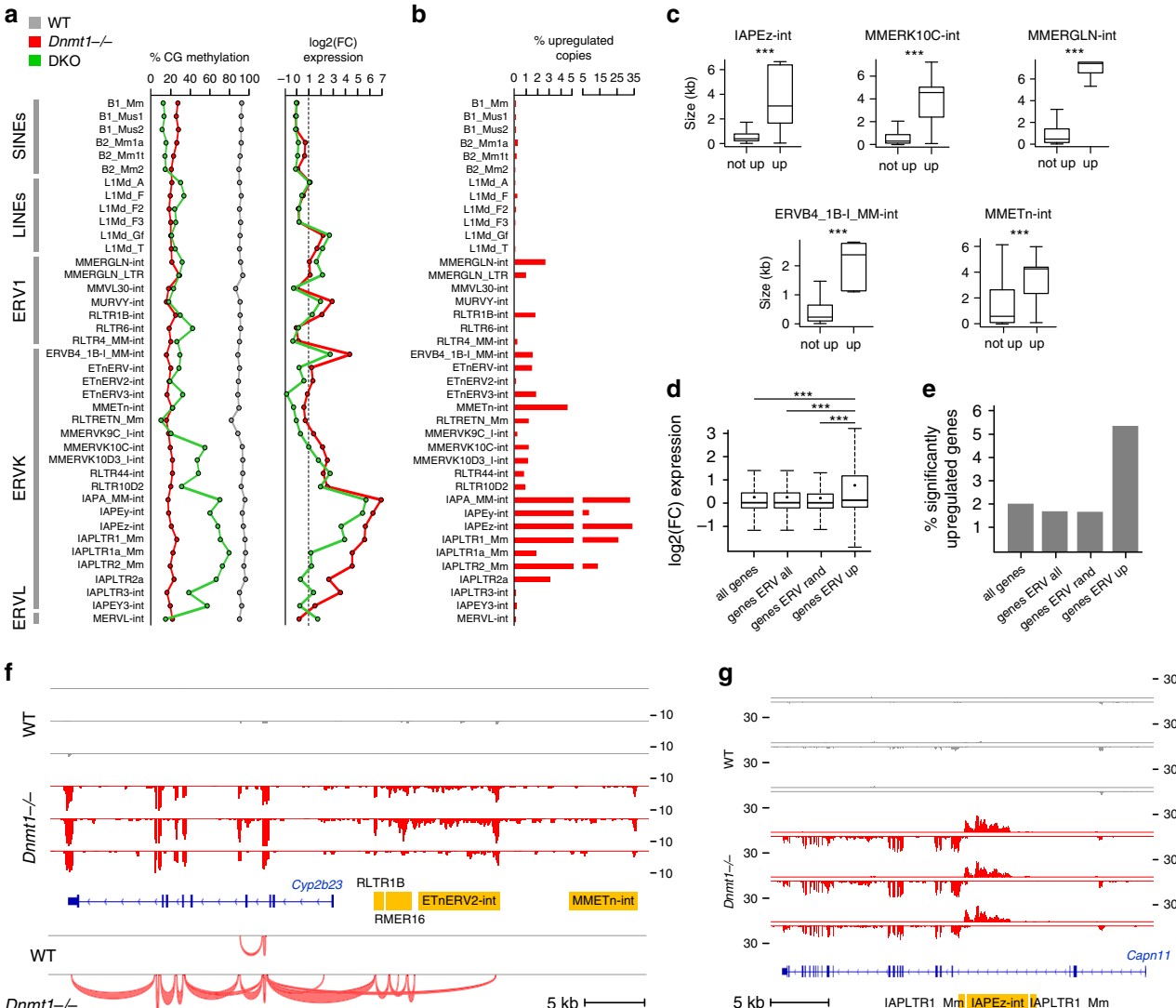

**Fig. 5 *Dnmt* mutant embryos show widespread transposon upregulation. a** DNA methylation measured by WGBS and fold change of expression of SINE, LINE, ERV1, ERVK, and EVRL retrotransposons in *Dnmt1*$^{-/-}$ and DKO embryos. Methylation and expression was calculated on RepeatMasker annotations. **b** Percentage of significantly upregulated copies within each retrotransposon family in *Dnmt1*$^{-/-}$ embryos. **c** Boxplots comparing the size of transposon copies upregulated in *Dnmt1*$^{-/-}$ embryos (up) or not upregulated (not up) for several ERV families. ***$p < 0.001$ (Wilcoxon test). In the boxplots the line indicates the median, the box limits indicate the upper and lower quartiles and the whiskers extend to 1.5 IQR from the quartiles. IAPEz-int: $n = 4835$ (not up), $n = 2484$ (up), $p = 0$; MMERVK10C-int: $n = 3193$ (not up), $n = 37$ (up), $p = 1.53e{-}14$; MMERGLN-int: $n = 834$ (not up), $n = 22$ (up), $p = 2.15e{-}12$; ERVB4_1B-I_MM-int: $n = 1347$ (not up), $n = 21$ (up), $p = 8.03e{-}12$; MMETn-int: $n = 936$ (not up), $n = 44$ (up), $p = 3.97e{-}14$. **d** Fold change of expression of genes located close to upregulated ERVs (genes ERV up, $n = 715$) compared with all genes, genes located close to all ERVs (genes ERV all) and a random selection of 715 genes located close to ERVs (genes ERV rand). In the boxplots the line indicates the median, the dot indicates the mean, the box limits indicate the upper and lower quartiles and the whiskers extend to 1.5 IQR from the quartiles. ***$p < 0.001$ (Wilcoxon test). **e** Percentage of significantly upregulated genes in *Dnmt1*$^{-/-}$ embryos for genes located close to upregulated ERVs and control genes. **f** *Cyp2b23* expression is induced by an upstream cluster of ERVs, which initiates a long RNA that splices into the exon 2. The figure shows RNA-seq tracks in WT and *Dnmt1*$^{-/-}$ embryos, along with splice junctions in one replicate of WT and *Dnmt1*$^{-/-}$ embryo. ERVs annotated by RepeatMasker are displayed in yellow. **g** RNA-seq tracks in WT and *Dnmt1*$^{-/-}$ embryos illustrating that the derepression of an intragenic IAPEz element leads to internal initiation of the *Capn11* gene on the opposite strand. The RNA-seq signals from the top (above the line) and bottom (below the line) strands are shown. IAPs annotated by RepeatMasker are displayed in yellow.

divisions in embryonic fibroblasts. Conversely, DNMT3A/B are strongly redundant and responsible for all de novo methylation in development, confirming what was speculated since the discovery of these enzymes[4], and excluding a de novo function of DNMT1 in embryonic development. This however does not exclude the possibility that DNMT1 catalyzes de novo methylation in other developmental contexts, for instance in oocytes[11,12]. Furthermore, we cannot exclude the possibility that some de novo activity of DNMT1 is implicated in the perdurance of DNA methylation in DKO embryos. Indeed, several families of ERVK

and LINE-1 retrotransposons are targets of TET enzymes in mouse embryonic cells[43], suggesting that a de novo activity of DNMT1 could be required to counteract TET-mediated demethylation at these sites.

The lack of evidence for a role of DNMT3A/B in maintenance methylation contradicts several studies showing that DNMT3A/B are required for maintenance methylation in ES cells[13–16,44]. One possible explanation for this discrepancy is the discovery that ES cells continuously cycle in and out of a transient hypomethylated state marked by MERVL expression[38]. Therefore the reduced

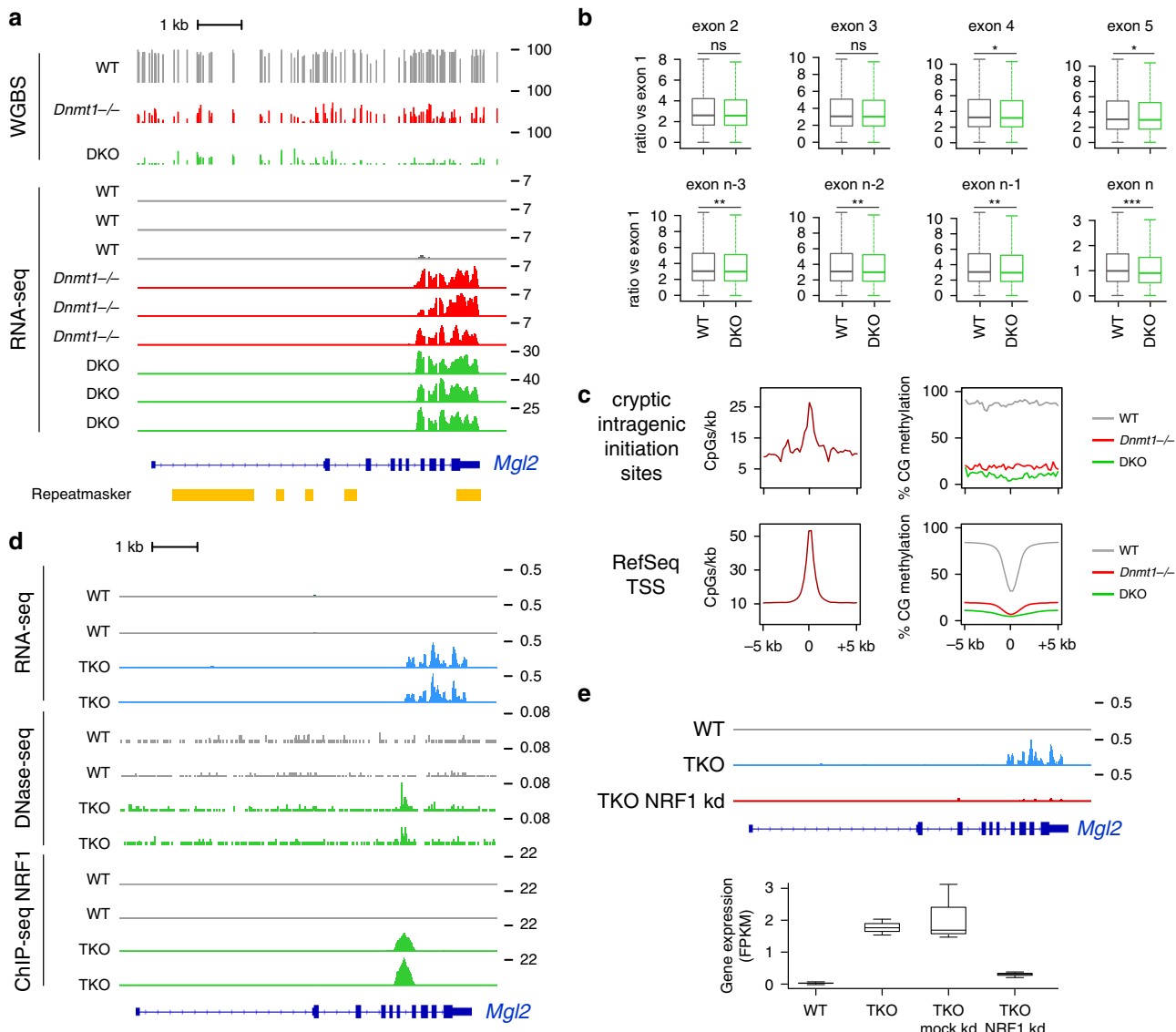

**Fig. 6 Transcripts initiate from cryptic intragenic promoters in *Dnmt* mutant embryos. a** Genome browser tracks of WGBS and RNA-seq profiles at the *Mgl2* locus in WT and *Dnmt* mutant embryos. One WGBS replicates and three RNA-seq replicates are shown. RepeatMasker annotations are displayed in yellow below the tracks. **b** Boxplot of the ratio of RNA-seq read counts in downstream exons compared to the first exon in WT and DKO embryos for all expressed genes with at least 5 exons ($n = 12,898$). Exon n represents the last exon of the gene. ns: not significant; *$p < 0.05$; **$p < 0.01$; ***$p < 0.001$ (Wilcoxon test). **c** Metaplots representing the CpG density and the CpG methylation levels in 5 kb sequences flanking cryptic intragenic initiation sites or canonical RefSeq TSS. **d** Genome browser tracks of RNA-seq, DNAse-seq and NRF1 ChIP-seq profiles at the *Mgl2* locus in WT and *Dnmt* triple knockout (TKO) ES cells. Two replicates are shown. **e** The genome browser tracks on the top display *Mgl2* RNA-seq profiles upon NRF1 knockdown in TKO ES cells. The boxplot on the bottom shows the quantification of *Mgl2* expression measured by RNA-seq (plotted as FPKM) upon mock and NRF1 knockdown in TKO ES cells ($n = 3$ replicates). In the boxplots the line indicates the median, the box limits indicate the upper and lower quartiles and the whiskers extend to 1.5 IQR from the quartiles.

methylation of *Dnmt3a/b* knockout ES cells could reflect a requirement for continuous de novo methylation to exit the MERVL+ state rather than true maintenance methylation. Our results also contradict a previous study that concluded on a role of DNMT3B in maintenance methylation in MEFs based on rough estimation of DNA methylation with restriction digestion[45]. In support of our conclusions, combined acute inactivation of DNMT3A/B does not lead to genome hypomethylation in human embryonic carcinoma cells[46].

Another key finding of our study is the interrogation of the transcriptional roles of DNA methylation by performing RNA-seq in severely hypomethylated embryos. We establish that DNA methylation of CpG-island promoters is a primary and causal

silencing mechanism for restricting ectopic expression of a large panel of germline genes. While we and others previously found some germline genes reactivated in partially hypomethylated embryos and cells[7,22,47–49], our study reveals that the set of germline genes repressed by CpG-island DNA methylation is larger than anticipated. This set includes many genes involved in the piRNA pathway, reinforcing the model that this might have evolved as a defense mechanism against transposons to couple genome demethylation with immediate activation of piRNA defense genes[47]. Future studies should be aimed at understanding the mechanisms that limit the expression of germline genes in preimplantation stages and subsequently direct de novo CpG-island DNA methylation to germline genes during development.

Our work reveals other functions of DNA methylation for gene regulation in embryos. Notably, we found that a small number of lineage-committed genes acquire promoter DNA methylation in WT embryos and are derepressed in methylation-deficient embryos. Furthermore the same genes display tissue-specific promoter hypomethylation in differentiated tissues. This strongly supports a role for DNA methylation in limiting precocious expression of lineage-committed genes in embryos. In addition, we demonstrate that intragenic methylation of CpG-rich sequences is essential to mask cryptic promoters in gene bodies and prevent the production of truncated gene transcripts. Using the *Mgl2* gene as a model, we were able to demonstrate that intragenic DNA methylation directly prevents methylation-sensitive transcription factors such as NRF1[24] from initiating cryptic intragenic transcripts. Previously, other epigenetic factors have been shown to limit cryptic intragenic initiation such as KDM5B and SETD2[50,51]. In addition, a recent study suggested that gene body DNA methylation suppresses widespread cryptic intragenic initiation in mouse ES cells[52]. In contrast to this report, we found no evidence for widespread intragenic transcription in hypomethylated embryos by quantifying RNA-seq signals in downstream versus the first exon of expressed genes. Instead, our results suggest that DNA methylation limits cryptic intragenic initiation from defined sequences in a small number of genes.

Another surprising finding is that 2C-genes are derepressed in DKO embryos. This was unexpected because DNA hypomethylation does not drive expression of 2C-genes in ES cells[38], which are instead repressed by CAF1, KDM1a, KAP1, G9a, HP1, and PRC1 in ES cells[39,53–56]. Recently it was found that *Dux*, *Dppa2*, and *Dppa4* activate the 2C program in 2-cell-like ES cells[57,58], although *Dux* has a minor role in activating genes in 2C embryos[59]. Interestingly *Dux*, *Dppa2,* and *Dppa4* are derepressed in DKO embryos (Supplementary Data 2), which could provide an explanation for the coordinated derepression of 2C-genes. Adding to the complexity, *Dux* and 2C-genes are not strongly activated in *Dnmt1*$^{-/-}$ embryos, suggesting either an indirect regulation by DNA methylation or a possible role of non-catalytic functions of DNMT3A/B. Hence, it is possible that DNMT3A/B repress 2C-genes by recruiting silencing complexes independently of their catalytic activity[60,61].

The prevailing model for transposon regulation is that they switch from H3K9me3-mediated silencing in preimplantation embryonic cells to a DNA methylation dominant mechanism in postimplantation embryos. However, the latter aspect of this model lacked experimental evidence in the mouse because, besides IAPs[17,29], it was unclear if other TE families require DNA methylation for repression. Our analysis demonstrates that DNA methylation is universally required to maintain repression of potentially active copies of numerous ERV and LINE transposons in postimplantation embryos, confirming that DNA methylation becomes a major epigenetic barrier against transposon expression in differentiated cells. Interestingly, SETDB1 is still required to repress some ERV1 transposons (MMVL30-int, RLTR6_Mm, and MULV-int) in mouse differentiated cells[62], which we find do not depend on DNA methylation.

In summary, our work provides a detailed description of the multiple functions of DNA methylation in maintaining the transcription integrity of mouse embryos. These results contribute to our understanding of why DNA methylation is essential for mammalian development.

## Methods

**Mouse lines and embryos**. All mouse lines used in the study were maintained on a C57BL/6J genetic background. All experimental animal procedures were performed following the ethical regulations of the Comité d'Ethique Régional en Expérimentation Animale de Strasbourg (CREMEAS). Mice were housed with free access to food and water, a 12 h light/dark cycle and controlled temperature (20–24 °C) and humidity (40–70%). We obtained a *Dnmt1*-null allele by crossing *Dnmt1*-2lox mice[30] with an ACTB-Cre deleter line[63], which creates a *Dnmt1* allele lacking the exons 4 and 5. *Dnmt1*$^{-/-}$ embryos were obtained by natural mating of heterozygous males and females. *Dnmt3a* and *Dnmt3b* knockout alleles were obtained by deleting critical catalytic exons as previously described[7]. We generated *Dnmt3a*$^{-/-}$ *Dnmt3b*$^{-/-}$ (DKO) embryos by natural mating of *Dnmt3a*$^{+/-}$ *Dnmt3b*$^{+/-}$ males and females. As controls, we recovered WT, *Dnmt3a*$^{+/-}$ and *Dnmt3a*$^{+/-}$ *Dnmt3b*$^{+/-}$ embryos from the same litters. The morning of the vaginal plug was designated E0.5 and embryos were manually dissected in M2 medium at E8.5. We simultaneously prepared genomic DNA and total RNA from the same embryos with the AllPrep DNA/RNA Mini Kit (Qiagen).

**Culture of MEFs and conditional inactivation of *Dnmts***. Mouse embryonic fibroblasts (MEFs) were isolated from E13.5 embryos and immortalized by serial passages. MEFs were grown in DMEM supplemented with 10% fetal bovine serum and 1% penicillin–streptomycin in a humidified atmosphere containing 5% CO2 at 37 °C. All cells were tested negative for mycoplasma contamination. *Dnmt3a*$^{2lox/2lox}$ *Dnmt3b*$^{2lox/2lox}$ MEFs were derived from an embryo obtained by crossing *Dnmt3a*-2lox and *Dnmt3b*-2lox mouse lines[45,64] on a C57BL/6J genetic background. *Dnmt1*$^{2lox/2lox}$ MEFs were derived from an embryo obtained by crossing *Dnmt1*-2lox mice[30] on a C57BL/6J genetic background. For conditional inactivation, the MEFs were transduced with a retrovirus coding for the Tamoxifen-inducible Cre-ERT2 recombinase and selected with puromycin (2 µg ml$^{-1}$). Recombination was induced by treating MEFs with 2 µM 4-OH-Tamoxifen (Sigma). The medium containing 4-OH-Tamoxifen was renewed every day during the first 2 weeks and then every 3 days. The efficiency of the recombination was validated by PCR genotyping, RT-qPCR, and western blotting. The conditional inactivation was performed 3 times independently and the cells were harvested at different time points of culture for genotyping and DNA methylation analysis by RRBS. The oligo sequences for PCR are provided in the Supplementary Data 4.

**Epigenetic editing with dCas9-TET1 fusion**. The plasmids coding for the dCas9-TET1 fusion were constructed based on the pdCas9-DNMT3A-EGFP plasmid (Addgene #71666). The EGFP sequence was substituted with the puromycin sequence from the PX459-V2 plasmid (Addgene #62988). Subsequently, the plasmid was digested with BamHI and FseI to replace the DNMT3A fragment with the catalytic domain of human TET1 (hTET1-CD). The sequence coding for hTET1-CD was synthesized by Integrated DNA Technologies (IDT) and amplified by PCR using forward and reverse primers introducing BamHI and FseI restriction sites. Two BbsI restriction sites within the TET1 sequence were removed by introducing silent mutations by site-directed mutagenesis. The gRNAs targeting *Dazl* (gRNA4: ACGCACTCCGTGGGCGACGT) and *Asz1* (gRNA5: GTGAAAGGCCAGCTCGTGGG) were designed using http://crispr.mit.edu, synthesized as pairs of oligonucleotides, annealed and cloned into the BbsI site. Immortalized MEFs isolated from a C57BL/6J embryo were transfected with the dCas9-hTET1-CD plasmid using Polyethylenimine (PEI) transfection reagent. In brief, 10 µg of plasmid and 20 µL of PEI were diluted in 250 µl of 150 mM NaCl each, combined and incubated for 30 min at RT. The complexes were added to 70–80% confluent MEF cells in 100 mm dishes. Twenty-four hours after transfection, the cells were selected with 3 µg mL$^{-1}$ of puromycin (Gibco, Thermo Fisher Scientific) for 48 h before harvesting the cells for DNA/RNA extraction using the AllPrep DNA/RNA Mini Kit (Qiagen).

**Epigenetic editing with the dCas9-Suntag-TET1 system**. The gRNAs were cloned in the pPlatTet-gRNA2 all-in-one vector[33]. pPlatTET-gRNA2 was a gift from Izuho Hatada (Addgene # 82559). Briefly, two 60mer oligonucleotides containing the gRNA sequence were annealed and extended to make a 100 bp double stranded DNA fragment using Q5 high fidelity polymerase (New England Biolabs #M0491S), and incorporated into the linearized pPlatTet-gRNA2 vector by Gibson assembly (New England Biolabs #E2611S). MEFs were transfected with the Neon electroporation system (Thermo Fisher Scientific) and cells expressing GFP were selected 72 h post-transfection by flow cytometry using a BD FACS Vantage cell Sorter (BD Biosciences) for DNA/RNA extraction using the AllPrep DNA/RNA Mini Kit (Qiagen).

**Gene expression analysis by RT-qPCR**. RNAs were reverse transcribed with the Maxima first strand cDNA synthesis kit (Thermo Fischer Scientific). qPCR was performed with the KAPA SYBR FAST qPCR kit (KAPA Biosystems) on a StepOnePlus PCR system (Applied Biosystems) using the standard curve method. We used fast PCR conditions as follows: 95 °C for 20 s, 40 cycles (95 °C for 20 s, 64 °C for 30 s), followed by a dissociation curve. The expression of target genes was normalized with three housekeeping genes (*Gusb*, *Rpl13a*, *B2m*, or *Mrpl32*). qPCR reactions were performed in triplicates with no-RT controls to rule out the presence of contaminating DNA. The oligo sequences are provided in the Supplementary Data 4.

**Western blot analysis**. Nuclear extracts were run on a SDS PAGE gel and transferred to a 0.2 µm nitrocellulose membrane. The membrane was blocked with

TBS, 0.1% Tween-20, 5% milk for 2 h at room temperature and incubated with the primary antibodies overnight at 4 °C. The membrane was washed three times, incubated with horseradish peroxidase-conjugated secondary antibody for 1 h at room temperature, and washed three times. The signal was detected by chemiluminescence using the ECL detection reagent (Amersham, GE Healthcare). The following primary antibodies were used: DNMT3A (NB120-13888, NovusBio, 1:200 dilution), LAMIN B1 (ab16048, Abcam, 1:2000 dilution).

**Bisulfite sequencing.** Hundred nanograms of genomic DNA were bisulfite converted using the EpiTect Bisulfite Kit (Qiagen) according to the manufacturer's instructions. The target regions were amplified by PCR with the Platinum Taq DNA Polymerase (Thermo Fisher Scientific) using the following conditions: 20 cycles of 30 s at 95 °C, 30 s at 58–48 °C (with a 0.5 °C decrease per cycle), 50 s at 72 °C followed by 35 cycles of 30 s at 95 °C, 30 s at 52 °C, 50 s at 72 °C. The PCR products were cloned by TA cloning in the pCR2.1 vector (TA Cloning Kit, Invitrogen) and 15–30 clones were sequenced. Sequences were aligned with the BISMA software and filtered to remove clonal biases. The oligo sequences are provided in the Supplementary Data 4.

**RRBS.** RRBS libraries were prepared from single embryos by *Msp*I digestion[7]. Fifty nanograms genomic DNA was digested for 5 h at 37 °C with *Msp*I (Thermo Fisher Scientific), end-repaired and A-tailed for 40 min at 37 °C with 5 U Klenow-fragment exo- (Thermo Fisher Scientific) and ligated to methylated adapters overnight at 16 °C with 30 U T4 DNA ligase (Thermo Fisher Scientific) in Tango 1X buffer. Fragments between 150 and 400 bp were excised from a 3% agarose 0.5X TBE gel, purified with the MinElute gel extraction kit (Qiagen) and bisulfite converted with the EpiTect bisulfite kit (Qiagen) with two consecutive rounds of conversion. Final libraries were amplified with PfU Turbo Cx hotstart DNA polymerase (Agilent) (2 min at 95 °C; 12–14 cycles of 30 s at 95 °C, 30 s at 65 °C, 45 s at 72 °C; final extension 7 min at 72 °C). Libraries were purified with AMPure magnetic beads (Beckman Coulter) and sequenced on an Illumina HiSeq4000 (2 × 75 bp) at Integragen SA (Evry, France). Reads were trimmed to remove low quality bases with Trim Galore v0.4.2 and aligned to the mm10 genome with BSMAP v2.74 (parameters -v 2 -w 100 -r 1 -x 400 -m 30 -D C-CGG -n 1). We calculated methylation scores using methratio.py in BSMAP v2.74 (parameters -z -u -g). Only CpGs covered by a minimum of eight reads were retained for analyses.

**WGBS.** WGBS was performed independently on two embryos for each genotype. Fifty nanograms of genomic DNA were fragmented to 350 bp using a Covaris E220 sonicator. DNA was bisulfite converted with the EZ DNA Methylation-Gold kit (Zymo Research) and WGBS libraries were prepared using the Accel-NGS Methyl-Seq DNA Library Kit (Swift Biosciences) according to the manufacturer's instructions with seven PCR cycles for the final amplification. The libraries were purified with Ampure XP beads (Beckman Coulter) and sequenced in paired-end (2 × 100 bp) on an Illumina HiSeq4000 at Integragen SA (Evry, France). Low quality bases as well as the first five bases of reads R1 and ten bases of reads R2 were trimmed with Trim Galore v0.4.2 (parameters -q 20 --clip_R1 5 --clip_R2 10). The reads were aligned to the mm10 genome and cleaned for duplicates using Bismark v0.22.1 with default parameters. Reads with signs of incomplete conversion were removed using the filter_non_conversion option in Bismark with the parameters --minimum_count 5 and --percentage_cutoff 50. Methylation calls were extracted using the Bismark methylation extractor. Only CpGs covered by a minimum of five reads were retained for analyses in each replicate. For global methylation analysis, both replicates of each genotype were combined by adding the number of Cs and Ts at each CpG position (mean sequencing depth WT: 24.81×; $Dnmt1^{-/-}$: 23.85×; DKO: 25.62×).

**Methylation data analysis.** The methylation of genomic features (Fig. 1f) was calculated by intersecting CpG positions with genomic annotations using the IRanges package in R, and averaging methylation of individual CpGs in each feature. We used the CpG-island annotation, RefSeq gene annotation, and RepeatMasker annotation (only elements > 400 bp) downloaded from the UCSC website, promoters were defined as −1 kb to +1 kb around RefSeq TSS, and intergenic regions were defined as genomic regions that do not overlap with any of the previous annotations. To identify regions with high residual methylation in ICM and DKO embryos, methylation was averaged in 1 kb windows containing at least 3 CpGs. To select regions de novo methylated during implantation (Fig. 2c), we selected 1 kb windows with <5% methylation in ICM and >50% methylation in E8.5 WT embryos. Metaplots of CG methylation in genes were generated by calculating methylation in twenty equal-sized windows within each RefSeq gene (excluding the X and Y chromosomes) and ten 1 kb windows of flanking sequences. Pairwise correlation plots of methylation scores were generated in 500 bp windows for RRBS and WGBS. Promoter classification based on CpG density was done as follows. For each promoter (−1 kb to +1 kb around RefSeq TSS), we calculated the CpG ratio and GC content in 500 bp sliding windows with 20 bp increments. LCP were defined as containing no window with a CpG ratio > 0.45, HCP were defined as containing at least one window with a CpG ratio > 0.65 and a GC content > 55%, and the remaining promoters were defined as ICP. To identify DMRs in *Dnmt3a/b* cDKO MEFs, we used eDMR from the methylKit R package with the following

criteria: at least three differentially methylated CpGs (DMCs), difference in methylation > 20%, adjusted *p* value < 0.001.

**RNA-seq and transcriptome analysis.** RNA-seq libraries were prepared from single embryos with the TruSeq Stranded Total RNA Sample Prep Kit with Ribo-Zero ribosomal RNA reduction (Illumina). We prepared libraries from three $Dnmt1^{-/-}$ and three WT littermate embryos, as well as six DKO, two WT, and four $Dnmt3a^{-/+}$ littermate embryos. The libraries were sequenced in paired-end (2 × 100 bp) on an Illumina HiSeq4000. Reads were mapped to the mm10 genome using TopHat v2.0.13 with a RefSeq transcriptome index and default parameters, reporting up to 20 alignments for multi-mapped reads. For data visualization, bigwig files were generated using bam2wig.py from the RSeQC package v2.6.4 (parameters -u -t 5000000000) and visualized in the Integrative Genomics Viewer (IGV). For differential gene expression analysis, reads mapping to repeats (with the exception of simple repeats and low complexity DNA sequences) were removed from the bam files using bedtools intersect to avoid that genes are falsely called upregulated because of the presence of TEs in UTRs. Unique reads were counted in RefSeq genes with HTSeq v0.7.2 (parameters –t exon –s reverse), and differentially expressed genes were determined using DESeq2 v1.16.1 (fold change > 3, adjusted *p* value < 0.001). For imprinted genes, the GTF annotation file was modified to allow quantification of *Snrpn*, *Gnas1a*, and *Nesp*. Splice junctions were visualized in IGV using the splice junctions files produced by TopHat. The FPKM values and PCA analysis were generated using DESeq2. Gene ontology analysis of differentially expressed genes was performed using DAVID 6.8 (https://david.ncifcrf.gov).

**Intragenic transcription.** To study intragenic transcription initiation, reads were counted in individual exons of all RefSeq gene isoforms using featureCounts from the Rsubread package v1.30.9, and the ratio of FPKM values of downstream exons over the first exon was plotted for all genes with at least 5 exons and a FPKM score > 1. To identify genes with intragenic initiation, differentially expressed exons were identified using DESeq2 v1.16.1 (fold change > 3, *p* value < 0.001) and the percentage of upregulated exon was calculated for each RefSeq isoform. The following criteria were then applied: (i) percentage of upregulated exons <100 %, (ii) the first exon is not upregulated, (iii) no other isoform of the same gene with 100% upregulated exons, (iv) if <5 upregulated exons they should be consecutive, (v) if ≥5 upregulated exons a gap of one exon is tolerated, (vi) the fold change of upregulated exons is >3 times higher than for the other exons, (vii) the FPKM values of upregulated exons is higher than for the other exons. Finally, the list of genes was manually curated to keep only one RefSeq isoform for each gene, eliminate false positives and eliminate genes initiating from transposons. For each gene, the position of intragenic initiation was defined manually from the bigwig files. Metaplots of CpG density and CpG methylation around cryptic intragenic initiation sites and RefSeq TSS were calculated in 250 bp windows.

**Transposon analysis.** Methylation of TE families was estimated by intersecting CpG positions with the UCSC RepeatMasker annotation using the IRanges package in R, and averaging the CpG methylation scores in each family. For the comparison of TE methylation between DKO embryos and ICM, only LTR and LINE families covered by at least 100 CpGs were retained. Expression of TEs was analyzed in several ways at the family and copy level. First, unique and multiple-mapping reads were counted in TE families using featureCounts from the Rsubread package v1.30.9 with a GTF file built from the UCSC RepeatMasker annotation, with the option to weight multi-mapping reads by the number of mapping sites (parameters countMultiMappingReads = TRUE, fraction = TRUE, useMetaFeatures = TRUE). In parallel, expression of TE families was also analyzed by mapping reads to RepBase consensus sequences using TopHat v2.0.13 allowing five mismatches, and counting reads with HTSeq v0.7.2. Differentially expressed TE families were identified using DESeq2 v1.16.1 (fold change > 2, adjusted *p* value < 0.001). To analyze the expression of individual copies of TEs, only unique reads were counted in individual TEs from the RepeatMasker GTF file using featureCounts (parameters countMultiMappingReads = FALSE, useMetaFeatures = FALSE). We then identified differentially expressed copies using DESeq2 v1.16.1 (fold change > 3, adjusted *p* value < 0.001), and calculated the percentage of upregulated copies within each TE family. The age of LINE-1 families was taken from Sookdeo et al.[65]. To represent sense and antisense transcription for IAPLTR1_Mm elements, elements were merged if distant from less than 8 kb. The RNA-seq signals were extracted in regions spanning from −5 kb to 5 kb from the start of the IAPLTR1_Mm elements on the forward and reserve strands using bwtool extract. The signals were averaged in 50 bp windows and plotted as metaplots or heatmaps using pheatmap in R.

**Datasets.** The following datasets were used: WGBS in gametes and early embryos (GSE56697), WGBS in mouse adult tissues (GSE42836), WGBS in B cells (GSE100262), WGBS in hematopoietic stem cells (GSE52709), RNA-seq in E3.5 ICM (GSE84234), RNA-seq, DNAse-seq, and NRF1 ChIP-seq in TKO ES cells (GSE67867).

**Statistics and reproducibility.** All measurements were biological replicates taken from individual embryos or independent experiments. Details on the statistical

tests and samples sizes are provided in the Figure legends. Statistical significance of differences in gene expression by RT-qPCR were evaluated by two-tailed unpaired Student's *t* test with unequal variances assuming normality of the distributions.

**Reporting summary**. Further information on research design is available in the Nature Research Reporting Summary linked to this article.

## Data availability

The methylome and RNA-seq sequencing data generated in this study are available in the NCBI Gene Expression Omnibus under the accession number GSE130735. All other data generated in this study are available in the Supplementary Information files or from the corresponding author upon reasonable request. The Source data for Figs. 1c, d, 3e, f, 4d–g, i, j and Supplementary Figs. 2c, 3a–c, h–j, 5e, and 8d are provided in the Source data file. The Source data for Figs. 3b, c, 4b, c, 5a, b and Supplementary Figs. 4c, d, h, 9a–f, and 11d are provided in the Supplementary Data files. UCSC genome annotations are available at http://genome.ucsc.edu. Source data are provided with this paper.

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

## Acknowledgements

We thank the staff of the IGBMC GenomEast sequencing platform and Annie Varrault for advices with the dCas9-Suntag-TET1 experiments. This work was funded by the European Research Council (ERC Consolidator grant no. 615371) and the Institut National du Cancer (INCa). T.D. was recipient of a Doctoral fellowship from the French Ministry for Higher Education and Research. A.A.L. was supported by the Fondation pour la Recherche Médicale (FRM).

## Author contributions

T.D. performed embryo dissection, conducted RRBS, WGBS, RNA-seq, and gene expression analysis and performed conditional inactivation experiments in MEFs. A.A.L. and H.A.A. performed dCas9-based methylation editing experiments. M.D. and A.F.B. conducted bioinformatic analysis of the sequencing data. A.B. performed embryo dissection and RRBS experiments. J.V. and R.P.N. contributed to conditional inactivation experiments in MEFs. M.T. and G.A. participated in the generation of mouse lines and embryo dissection. M.W. conceived and supervised the project and performed data analysis. M.W. and T.D. wrote the paper with contributions from the other authors.

## Competing interests

The authors declare no competing interests.
