## [Peer Review File · Nature Communications]

Reviewers' comments:

Reviewer #1 (Remarks to the Author):

Synopsis

In this manuscript, Dahlet et al. present a thorough and well executed analysis of the roles played by the DNA methyltransferases DNMT1, DNMT3A, and DNMT3B in the establishment and maintenance of DNA methylation patterns during early mouse development. Using well-established genetic models, state-of-the-art genome-wide profiling techniques, and high-quality data, their work presents a number of key findings contributing to our understanding of the roles played by these enzymes during development, with an emphasis on the transition from a highly hypomethylated inner cell mass stage to early post-implantation stages where widespread de novo DNA methylation has occurred. First, they confirmed by RRBS and WGBS that DNMT1 plays an essential role in DNA methylation levels in early embryos. They also demonstrated the essential roles of DNMT3A and 3B in de novo DNA methylation in vivo and discovered that embryos deficient for those two enzymes (DKO) show a DNA methylation profile very similar to the one observed in inner cell mass cells of the blastocysts, prior to the main wave of de novo methylation, highlighting the role played by DNMT1-catalyzed maintenance. The absence of maintenance roles for the DNMT3 enzymes was also shown via acute deletions generated in MEFs. Another important finding is to expand on the previously described function of DNA methylation in repressing germ cell genes in somatic cells. The functional importance of 5mC in this repressive function was also nicely addressed via a CRISPR-targeted TET1 approach in MEFs. Similar functions were uncovered for 2-cell stage transcripts, specific families of transposable elements, cryptic intragenic promoters, and cell type-specific genes.

Overall the manuscript is very well written and the presentation of the results is excellent. Although much of the data presented confirms already described functions for these enzymes, this is the first detailed analysis of those functions in vivo, in E8.5 embryos, at an early postimplantation stage following a major wave of "genome-wide" de novo DNA methylation. The fact that both similarities and differences were noted with previously published datasets from wild-type and mutant mouse ESCs, reinforces the importance of studying these processes in vivo, within their natural context. To this reviewer, this is one of the main strength of this elegant study, one which is likely to assure its broad interest to the readership of Nature

Communications. The definitive nature of some of the data and analyses presented will ensure the continued referencing to their key findings, which have broad impact on our understanding of such areas of research as developmental regulation, differentiation, protection of the somatic program, repression of transposable elements, and genomic imprinting.

Major comments

1. p. 4, line 9 - This statement is too general and needs to be qualified. Although the results presented convincingly show that together, DNMT3A and DNMT3B are responsible for the bulk is not all de novo DNA methylation occurring between the blastocyst and E8.5 stages, the possibility that DNMT1 alone can catalyze de novo DNA methylation in other developmental contexts, for instance in oocytes, cannot be excluded from this data.

2. Figure 3C - What about imprinted genes regulated by maternal H3K27me3, such as *Sfmbt2*, *Gab1* and *Phf17*? See Inoue et al. 2017. Since this oocyte-derived repressive mark appears to be specifically maintained in the trophoblast lineage, and lost in the epiblast, the data presented here cannot unfortunately confirm whether or not the paternal-allele specific expression of those imprinted genes is indeed DNA methylation-independent. Do the authors have E8.5 extra-embryonic tissues they could use to look at this aspect by RT- qPCR for those genes?

3. p.5, line 9 - Based on their DNA methylation profiling, the authors conclude that DNMT1 cannot

"methylate previously unmethylated sequences in embryos." In light of recent evidence suggesting the ability of DNMT1 to use unmethylated CpGs as substrates, for instance in Stella-deficient oocytes, how can the results presented formally eliminate the possibility that at least some of the residual 5mC seen in DKO embryos, which, as the authors rightly point out, is similar to the 5mC profile in ICM cells, is not dependent on some de novo activity of DNMT1? For instance, could such an activity be implicated in the perdurance of the DNA methylation of IAPs in DKO embryos? Are some of the methylated regions observed in DKO embryos known target for TET enzymes, which could support this model implicating an embryonic de novo methylating activity for DNMT1?

4. p. 5, line 30 - In this respect, the line stating that "all imprinted loci" are regulated by DNA methylation is also an over-generalization. Have the authors indeed look at ALL imprinted genes? Furthermore, this relationship might not apply to H3K27me3-regulated imprinted genes in the extra-embryonic trophoblast lineage.

5. In Supplementary Table 2, the imprinted genes have not been colour-coded as indicated in the legend, with the exception of Zdbf2.

6. p. 8, line 28 - At the end of this section, the authors suggest an essential role for de novo DNA methylation in silencing the 2-cell stage program in post-implantation embryos. However, as they note above, the derepression of this program, including both clustered single-copy genes and specific ERV families, is not observed in Dnmt1^{-/-} E8.5 embryos. How is this observation consistent with the proposed repressive role for de novo DNA methylation? I agree with the conclusion that the effects of DNA methylation here is most likely indirect, perhaps via the ectopic expression of DUX/DPPA2/DPPA4 in DKO embryos, but isn't the same expected to also occur in Dnmt1^{-/-} embryos?

7. One of the major limitations of the current study is the strictly DNA methylation-centric view of the analysis presented. DNA methylation does not work in isolation in mammals and several different examples of interplay with histone modifications have been described in ESCs and in vivo. Regarding the function of DNMT3A and 3B, an important area of research has been the description of mechanisms implicated in the genomic targeting of those enzymes, thus defining their sites of action and the resulting genomic DNA methylation patterns. One such mechanism is via the transcription-coupled deposition of H3K36me3 by SETD2 in differentiating ESCs (Baubec, Nature, 2015) and in oocytes (Xu, Nature Genetics, 2019). Since Dahlet et al. have now described in details the sites of de novo DNA methylation by the combined actions of DNMT3A and 3B, they could intersect their data with H3K36me3 profiling in similar stage embryos and draw important conclusions of the molecular rules guiding their activities in vivo: i) What fraction of de novo methylated CpGs are within H3K36me3-marked transcribed regions? ii) What other categories of non-transcribed genomic regions, if any, are also targeted for methylation in vivo? In turn, this additional data could open new future avenues to explore the basic rules guiding the activity of those methyltransferases. If the authors have access to the necessary H3K36me3 ChIP-seq profiles, the proposed analysis should be added to the manuscript.

Minor comments

A. p.9, line 2 - The ectopic activation of IAP in Dnmt1^{-/-} embryos was shown by northern blot and ISH by the authors of the paper cited here.

B. p.12, line 26 - Although the authors cite the DPPA papers, they do not mention the recent work on DUX by Chen & Zhang (Nature, 2019). A brief statement summarizing the key findings of this recent publication should be included.

C. p. 13, line 13 - Change "We created..." to "We obtained...", since this allele has previously been engineered and published, and was not "created" in this study.

D. p.13, line 16 - Change "...were created..." to "...were obtained..."

E. p. 13, line 27 - "...negative for Mycoplasma contamination".

F. Supplementary Tables. All gene names should be in italics.

G. p. 14, line 20 - The drug should be "puromycin" not "puromycine" .

H. Fig. 1A - Add a label for the stage presented (E8.5) to panel itself, not only in the legend.

I. Fig. 1B - "Average distribution OF..."

J. Fig. 2B - Add the coefficient of correlation to the plot.

K. Fig. 3B - The list of all the imprinted germline DMRs analyzed must be provided, perhaps in an additional Supplementary Table.

Reviewer #2 (Remarks to the Author):

The paper by Dahlet et al. describes a comprehensive and paired genome wide DNA-methylation and expression comparison of Dnmt1^{-/-} and Dnmt3A/Dnmt3B double knockouts (DKO) at d8.5 using RRBS and mRNA-Seq data. The authors analyse tissues of e8.5 mice lacking Dnmt1 or Dnmt3a/b (DKO). These mice die around this stage with severe developmental and growth defects. This has been described long time ago, while the precise epigenetic deficits were only partially analysed. The present analysis aims at filling this gap. The authors performed a genome wide analysis to determine the consequences for a loss of Dnmt1 and Dnmt3A/B, respectively, for maintenance and de novo methylation, the control of developmental genes and the control of transposable elements. Overall the paper confirms a number of previous observations reported by others mostly from „pre“-epigenomics times. The findings support the general assignment of Dnmt functions and add a few new interesting aspects on TE and germ line control. The author confirm i) Dnmt1's main role for maintenance during postimplantation development and the general maintenance of imprinted gene methylation at ICRs/gDMRs, ii) the methylation dependent repression of distinct classes of transposable elements by either Dnmt3a/b or Dnmt1 (partially new) and iii) the postimplantation regulation of lineage-committed genes by Dnmt3A/B and germ line active genes by Dnmt1 (partially new). The paper comprises a series of data analyses complemented by the use of external datasets for sperm, oocyte, ICM, 2C, 4C and 2Clike ES cell. While the authors filter out a fair amount of interesting (mainly confirmatory) data, some aspects on the control of transposable elements and the link between DKO and Mervl driven 2C like gene expression as well as the suppression of germ line programs adds some new points to the discussion. On the other hand the setup of the experiments and their documentation raise several general and specific (also minor) concerns:

General:

1) The authors extract DNA and RNA from total Dnmt1^{-/-} and Dnmt3a/b DKO's embryos at e8.5 and

compare their expression and methylation profiles. At this timepoint embryos show similar but certainly not identical phenotypes. This causes a (uncontrollable) variation as tissue composition and developmental staging between the embryos and hence confound the expression as well as DNA-methylation comparisons. There are many open questions: One concerns the unclear point how many replicates have been included in RNA-Seq analyses – how divergent were these – there is one short statements (page 6 line 13/14) with „data not shown“. Ultimately single cell RNA profiling would probably be needed to understand the developmental differences/similarities caused by different origins of hypomethylation.

2) The incredible high correlation of RRBS and WGBS data (Supplementary Figure 1B, C, E) stands in stark contrast to the difference visible in Figure 1B and 1E particularly around the TSS and gene body in WT and in KO's. The overall levels of methylation measured by WGBS differ between 10% and 20% for both CGI's and non CGI's in Dnmt1^{-/-} and DKO's (slightly less). Which filtered data are used for the genome wide comparison? I have never seen a correlation of 1 in any type of RRBS or WGBS datasets.

3) The (replicated) RRBS data shown in Figure 1 are mainly used to „validate“ the WGBS data (for which it is unspecified if they are based on replicates as well). Following a first general genome wide comparison (including the RRBS replicates) all subsequent analyses are performed with single (?) WGBS datasets for WT, Dnmt1^{-/-}, DKO's hence analysing CpGs at 5x coverage across all samples – this implies a minimum error range of 20% at individual positions! All other datasets important for interpretations were imported from other (previous) studies.

4) It remains unclear why Dnmt1^{-/-} is not included in comparison to DKO in Figure 2.

5) The experiments shown to support the lack of maintenance contribution of Dnmt3a/b in MEFs is unclear. There are some technical concerns: The authors do neither show western blot data or RNA seq data to confirm the timing of Dnmt3a and 3b loss– nor do they comment on effects of long term passaging (do they get possible selection effects when primary MEFs get into crisis). Moreover the experiment does not provide a real information on Dnmt3a/b contribution of maintenance. The conclusion is that there is no loss of overall methylation! An induced KO of Dnmt1 would have been the real control to show to which extent Dnmt3a/b contribute to „maintain methylation“.

Minor:

The authors do not explain how RRBS libraries were constructed (by MSpI digestion?) and if this explains (biased) differences between WGBS and RRBS.

The reason why the author speak about „heterochromatic“ clusters in the text referring to Suppl Figure 8D remains totally unclear. There are no data showing the heterochromatin connection.

The nice examples of the sashimi plots in supple figure 10 e.g. for Cyp2b23 should be included into the main figure figure 6.

The authors state in their discussion that the results provide „an important clarification on the function of“ DNMT's – most of functional aspects have been known /discussed before and are „comprehensively validated“ but certainly not clarified.

Reviewer #3 (Remarks to the Author):

For about two decades, it has been known that DNA methyltransferases, DNMT1, DNMT3A, and

DNMT3b, are essential for mouse embryonic development. In this manuscript, Dahlet et al. report the first genome-wide methylation analyses on Dnmt1 KO and Dnmt3ab DKO embryos, which revealed that Dnmt3a and Dnmt3b are together required for de novo methylation after implantation whereas Dnmt1 is solely required for maintenance methylation. They also revealed that DNA methylation is important to repress a number of germline genes as well as many retrotransposons such as IAP, L1, and MERVK10C. Their findings also include that DNA methylation plays a role in suppressing cryptic transcription in gene bodies, and that CpG-rich promoters of lineage-committed genes once gain DNA methylation after the genome-wide erasure of methylation in ICM, and this methylation is important to repress these genes in unrelated tissues. The manuscript is generally well written and data are presented appropriately. However, most of the results seem to be an extension of the previous knowledge (reported before the NGS era), which declines the conceptual novelty of the manuscript. I have some concerns listed below.

- (1) For expression analyses, RNAs were prepared by ribosome depletion (Ribo-zero). Thus, pre-mRNAs were included in the RNA preparation. Because many TEs reside in introns, polyA(+) RNA purification is better to quantify the TE expression. Alternatively, the authors should provide some data supporting that the existence of pre-mRNAs does not have a significant impact.
- (2) To identify unregulated genes by RNA-seq, the sequence reads that were mapped on to genomic regions annotated as a TE should be removed from the bam files after Tophat mapping, because they could be incorrectly mapped onto TE sequences present in UTRs of genes and this could give false upregulation if the same family of TE copies somewhere in the genome are upregulated.
- (3) In page 9, lines 8 to 18 (results shown in Fig. 5B), the authors counted upregulated TE copies in the genome by using uniquely mapped reads. It should be mentioned that this method underestimates it, because very young TE copies (which likely retain the transcriptional potential) cannot be uniquely mapped. Less than 0.1% of L1Md_A copies showed upregulation. But when analyzed in bulk as shown in Fig. S9C, L1Md_A showed about 4-fold upregulation. Same for L1Md_Tf.
- (4) In page 9, middle, the authors identified TEs that drive expression of nearby genes, similar to the situation of Setdb1 KO ES cells (ref.11). It should also be mentioned that loss of DNA methylation and TE upregulation leads to the upregulation of nearby genes in spermatogenic cells (Inoue et al. PLoS Genet 2017, 13: e1006926).
- (5) In page 9, lines 25 to 27, it should be mentioned that the IAP sequences are inserted in the antisense orientation to the host genes, Trpm2 and Capn11. When analyzing TE expression using the rebase sequences, did the authors see upregulation of the antisense transcription of IAPLTR1_Mm?
- (6) In the previous report (Inoue et al. PLoS Genet 2017), loss of methylation upregulates the antisense L1 promoters. This reviewer wonders if this was also observed in the Dnmt1 KO and Dnmt3ab DKO embryos.
- (7) Related to the section in page 8, "de novo methylation is required to repress 2C-specific genes", it is possible that the presence of DNMT3A and/or DNMT3B proteins is required for the repression. For example, they may bind and recruit a silencing complex to these promoters. This can be tested by using the catalytically inactive mutants, which can be left for future studies. But, at least, it is better to discuss this possibility.
- (8) In page 10, lines 13-15, the authors stated the sites of cryptic transcription start are methylated in WT embryos. Did they see loss of methylation in the Dnmt mutants? The statistical data for the mutants should be presented in Fig. 6C along with the WT data. Related to this, whereas only the example of Mgl2 is shown in Fig. 6AD, a few other examples can be included as supplementary figures to demonstrate the generality.
- (9) In page 4, line 11, "...occurs uniformly across all genomic sequences...". As far as inspected for Fig. 4G (the Dnmt1^{-/-} track) and Fig.1I, it does not look uniform. Regions of >80% methylation in wild-type showed 10-50% methylation remained in the Dnmt1 KO. Related to this, it is better to include a Dnmt1 KO track in Fig. 2A to show genomic regions with high methylation levels in E3.5 ICM and E8.5 DKO indeed lost methylation in E8.5 Dnmt1 KO.

(10) Page 4 line 34 and Page 6, line 19. The authors stated “strong positive correlation” and “good correlation” respectively. Although these correlations can be seen in Fig. 2B and Fig. S4D, the R values should be provided.

(11) In Fig. H, it is unclear from the main text and legend whether single CpG sites were analyzed or single bisulfite reads were analyzed.

(12) In Fig. S5AC, please indicate the methylation levels (%). In Figs. 4E and S5E, please indicate the p-values, as the *Dazl* expression with gRNA showed a large experimental variation.

We thank the reviewers for their positive evaluation of our work and their constructive criticisms and suggestions that greatly helped to improve the manuscript. We performed a number of additional experiments, analyses and text edits in response to these comments, which are detailed in the point-by-point response below.

Reviewer #1 (Remarks to the Author):

In this manuscript, Dahlet et al. present a thorough and well executed analysis of the roles played by the DNA methyltransferases DNMT1, DNMT3A, and DNMT3B in the establishment and maintenance of DNA methylation patterns during early mouse development. Using well-established genetic models, state-of-the-art genome-wide profiling techniques, and high-quality data, their work presents a number of key findings contributing to our understanding of the roles played by these enzymes during development, with an emphasis on the transition from a highly hypomethylated inner cell mass stage to early post-implantation stages where widespread de novo DNA methylation has occurred. First, they confirmed by RRBS and WGBS that DNMT1 plays an essential role in DNA methylation levels in early embryos. They also demonstrated the essential roles of DNMT3A and 3B in de novo DNA methylation in vivo and discovered that embryos deficient for those two enzymes (DKO) show a DNA methylation profile very similar to the one observed in inner cell mass cells of the blastocysts, prior to the main wave of de novo methylation, highlighting the role played by DNMT1-catalyzed maintenance. The absence of maintenance roles for the DNMT3 enzymes was also shown via acute deletions generated in MEFs. Another important finding is to expand on the previously described function of DNA methylation in repressing germ cell genes in somatic cells. The functional importance of 5mC in this repressive function was also nicely addressed via a CRISPR-targeted TET1 approach in MEFs. Similar functions were uncovered for 2-cell stage transcripts, specific families of transposable elements, cryptic intragenic promoters, and cell type-specific genes.

Overall the manuscript is very well written and the presentation of the results is excellent. Although much of the data presented confirms already described functions for these enzymes, this is the first detailed analysis of those functions in vivo, in E8.5 embryos, at an early postimplantation stage following a major wave of “genome-wide” de novo DNA methylation. The fact that both similarities and differences were noted with previously published datasets from wild-type and mutant mouse ESCs, reinforces the importance of studying these processes in vivo, within their natural context. To this reviewer, this is one of the main strength of this elegant study, one which is likely to assure its broad interest to the readership of Nature Communications. The definitive nature of some of the data and analyses presented will ensure the continued referencing to their key findings, which have broad impact on our understanding of such areas of research as developmental regulation, differentiation, protection of the somatic program, repression of transposable elements, and genomic imprinting.

Major comments

1. p. 4, line 9 - This statement is too general and needs to be qualified. Although the results presented convincingly show that together, DNMT3A and DNMT3B are responsible for the bulk is not all de novo DNA methylation occurring between the blastocyst and E8.5 stages, the possibility that DNMT1 alone can catalyze de novo DNA methylation in other developmental contexts, for instance in oocytes, cannot be excluded from this data.

We agree with the reviewer that our statement was too general and we changed it as follows: *"Altogether this indicates that DNMT3A/B are responsible for the bulk de novo DNA methylation in embryos between the blastocyst and E8.5 stages and that DNMT1 has a negligible capacity for de novo DNA methylation during embryonic development"*.

In addition we added the following sentence in the discussion: "*This however does not exclude the possibility that DNMT1 can catalyze de novo DNA methylation in other developmental contexts, for instance in oocytes (refs 11-12)*".

2. Figure 3C - What about imprinted genes regulated by maternal H3K27me3, such as Sfbmt2, Gab1 and Phf17? See Inoue et al. 2017. Since this oocyte-derived repressive mark appears to be specifically maintained in the trophoblast lineage, and lost in the epiblast, the data presented here cannot unfortunately confirm whether or not the paternal-allele specific expression of those imprinted genes is indeed DNA methylation-independent. Do the authors have E8.5 extra-embryonic tissues they could use to look at this aspect by RT- qPCR for those genes?

We agree with the reviewer that this is a very interesting point, however we were unable to perform this experiment for two reasons. First, to avoid maternal cell contamination, this would require to use a GFP reporter allele as described in Inoue et al. 2017. Second, to address this question carefully, it is crucial to have allelic information by using reciprocal hybrid embryos as described in Inoue et al. 2017 or Hanna et al. 2019. Finally, the role of DNA methylation in the maintenance of non-canonical imprinting in extraembryonic tissues has been recently addressed in a publication from the Zhang laboratory: Chen Z et al., Sci Adv. 2019 5(12):eaay7246.

3. p.5, line 9 - Based on their DNA methylation profiling, the authors conclude that DNMT1 cannot "methylate previously unmethylated sequences in embryos." In light of recent evidence suggesting the ability of DNMT1 to use unmethylated CpGs as substrates, for instance in Stella-deficient oocytes, how can the results presented formally eliminate the possibility that at least some of the residual 5mC seen in DKO embryos, which, as the authors rightly point out, is similar to the 5mC profile in ICM cells, is not dependent on some de novo activity of DNMT1? For instance, could such an activity be implicated in the perdurance of the DNA methylation of IAPs in DKO embryos? Are some of the methylated regions observed in DKO embryos known target for TET enzymes, which could support this model implicating an embryonic de novo methylating activity for DNMT1?

We agree with the reviewer that we cannot formally distinguish between maintenance and *de novo* activity of DNMT1 at sites that retain DNA methylation in DKO embryos. Indeed, it is true that some families of ERVK and LINE-1 retrotransposons are known targets of TET enzymes in mouse ES cells (de la Rica L et al. 2016, Genome Biol 17:234), which could indicate that some *de novo* activity of DNMT1 is required to counteract TET activity. Some experiments could be envisioned to address this experimentally for example by combining *Dnmt* and *Tet* mutations in embryos, which we were unable to do. Nevertheless we thank the reviewer for this comment and added the following text in the discussion:

"Furthermore, we cannot formally exclude the possibility that some de novo activity of DNMT1 is implicated in the perdurance of DNA methylation in DKO embryos. Indeed, several families of ERVK and LINE-1 retrotransposons are known targets of TET enzymes in mouse embryonic cells (ref 43), suggesting that a de novo activity of DNMT1 could be required to counteract TET-mediated demethylation at these sites."

4. p. 5, line 30 - In this respect, the line stating that "all imprinted loci" are regulated by DNA methylation is also an over-generalization. Have the authors indeed look at ALL imprinted genes? Furthermore, this relationship might not apply to H3K27me3-regulated imprinted genes in the extra-embryonic trophoblast lineage.

We agree that "all imprinted loci" was an over-generalization and we changed the sentence as follows: "*provide in vivo validation for the role of DNMT1 at many DNA methylation-dependent imprinted loci*".

5. In Supplementary Table 2, the imprinted genes have not been colour-coded as indicated in the legend, with the exception of Zdbf2.

We wished to highlight only deregulated imprinted genes in purple. The color legend in the table has been clarified.

6. p. 8, line 28 - At the end of this section, the authors suggest an essential role for de novo DNA methylation in silencing the 2-cell stage program in post-implantation embryos. However, as they note above, the derepression of this program, including both clustered single-copy genes and specific ERV families, is not observed in *Dnmt1*^{-/-} E8.5 embryos. How is this observation consistent with the proposed repressive role for de novo DNA methylation? I agree with the conclusion that the effects of DNA methylation here is most likely indirect, perhaps via the ectopic expression of DUX/DPPA2/DPPA4 in DKO embryos, but isn't the same expected to also occur in *Dnmt1*^{-/-} embryos?

We agree with the reviewer's concern about the role of *de novo* DNA methylation in the silencing of 2C genes. The fact that 2C and MERVL transposons are not strongly upregulated in *Dnmt1*^{-/-} embryos is a strong argument against a direct role of *de novo* methylation. Therefore we changed the title of the paragraph from "*De novo DNA methylation is required to repress 2C-specific genes*" to "*De novo DNA methyltransferases are required to repress 2C-specific genes*".

Furthermore, we expanded the discussion to insist on possible indirect roles of DNA methylation or non-catalytic functions of DNMT3A/B enzymes. The following text was added in the discussion: "*Adding to the complexity, Dux and 2C genes are not strongly activated despite hypomethylation in Dnmt1^{-/-} embryos, suggesting either an indirect regulation by DNA methylation or a possible role of non-catalytic functions of DNMT3A/B. Hence, it is possible that DNMT3A/B repress 2C genes by recruiting silencing complexes to these promoters independently of their catalytic activity (ref 61,62).*"

7. One of the major limitations of the current study is the strictly DNA methylation-centric view of the analysis presented. DNA methylation does not work in isolation in mammals and several different examples of interplay with histone modifications have been described in ESCs and in vivo. Regarding the function of DNMT3A and 3B, an important area of research has been the description of mechanisms implicated in the genomic targeting of those enzymes, thus defining their sites of action and the resulting genomic DNA methylation patterns. One such mechanism is via the transcription-coupled deposition of H3K36me3 by SETD2 in differentiating ESCs (Baubec, Nature, 2015) and in oocytes (Xu, Nature Genetics, 2019). Since Dahlet et al. have now described in details the sites of de novo DNA methylation by the combined actions of DNMT3A and 3B, they could intersect their data with H3K36me3 profiling in similar stage embryos and draw important conclusions of the molecular rules guiding their activities in vivo: i) What fraction of de novo methylated CpGs are within H3K36me3-marked transcribed regions? ii) What other categories of non-transcribed genomic regions, if any, are also targeted for methylation in vivo? In turn, this additional data could open new future avenues to explore the basic rules guiding the activity of those methyltransferases. If the authors have access to the necessary H3K36me3 ChIP-seq profiles, the proposed analysis should be added to the manuscript.

We agree with the reviewer that this would be a very interesting analysis. Unfortunately, after a thorough mining of available ChIP-seq data, we were unable to find H3K36me3 ChIP-seq datasets from similar stage embryos.

Minor comments

A. p.9, line 2 - The ectopic activation of IAP in *Dnmt1*^{-/-} embryos was shown by northern blot and ISH by the authors of the paper cited here.

B. p.12, line 26 - Although the authors cite the DPPA papers, they do not mention the recent work on DUX by Chen & Zhang (Nature, 2019). A brief statement summarizing the key findings of this recent publication should be included.

C. p. 13, line 13 - Change “We created...” to “We obtained...”, since this allele has previously been engineered and published, and was not “created” in this study.

D. p.13, line 16 - Change “...were created...” to “...were obtained...”

E. p. 13, line 27 - “...negative for Mycoplasma contamination”.

F. Supplementary Tables. All gene names should be in italics.

G. p. 14, line 20 - The drug should be “puromycin” not “puromycine” .

H. Fig. 1A - Add a label for the stage presented (E8.5) to panel itself, not only in the legend.

I. Fig. 1B - “Average distribution OF...”

We thank the reviewer for these suggestions (A-I), which have all been corrected in the revised manuscript.

J. Fig. 2B - Add the coefficient of correlation to the plot.

The coefficient of correlation ($r= 0.80$) was added to the plot in the **Figure 2b.**

K. Fig. 3B - The list of all the imprinted germline DMRs analyzed must be provided, perhaps in an additional Supplementary Table.

We thank the reviewer for this suggestion and now provide a list of gDMRs with their methylation values in the **Supplementary Data 1.**

Reviewer #2 (Remarks to the Author):

The paper by Dahlet et al. describes a comprehensive and paired genome wide DNA-methylation and expression comparison of Dnmt1-/- and Dnmt3A/Dnmt3B double knockouts (DKO) at d8.5 using RRBS and mRNA-Seq data. The authors analyse tissues of e8.5 mice lacking Dnmt1 or Dnmt3a/b (DKO). These mice die around this stage with severe developmental and growth defects. This has been described long time ago, while the precise epigenetic deficits were only partially analysed. The present analysis aims at filling this gap. The authors performed a genome wide analysis to determine the consequences for a loss of Dnmt1 and Dnmt3A/B, respectively, for maintenance and de novo methylation, the control of developmental genes and the control of transposable elements.

Overall the paper confirms a number of previous observations reported by others mostly from „pre“-epigenomics times. The findings support the general assignment of Dnmt functions and add a few new interesting aspects on TE and germ line control. The author confirm i) Dnmt1’s main role for maintenance during postimplantation development and the general maintenance of imprinted gene methylation at ICRs/gDMRs, ii) the methylation dependent repression of distinct classes of transposable elements by either Dnmt3a/b or Dnmt1 (partially new) and iii) the postimplantation regulation of lineage-committed genes by Dnmt3A/B and germ line active genes by Dnmt1 (partially new).

The paper comprises a series of data analyses complemented by the use of external datasets for sperm, oocyte, ICM, 2C, 4C and 2Clike ES cell. While the authors filter out a fair amount of interesting (mainly confirmatory) data, some aspects on the control of transposable elements and the link between DKO and Mervl driven 2C like gene expression as well as the suppression of germ line programs adds some new points to the discussion. On the other hand the setup of the experiments and their documentation raise several general and specific (also minor) concerns:

General:

1) The authors extract DNA and RNA from total *Dnmt1*^{-/-} and *Dnmt3a/b* DKO's embryos at e8.5 and compare their expression and methylation profiles. At this timepoint embryos show similar but certainly not identical phenotypes. This causes a (uncontrollable) variation as tissue composition and developmental staging between the embryos and hence confound the expression as well as DNA-methylation comparisons. There are many open questions: One concerns the unclear point how many replicates have been included in RNA-Seq analyses – how divergent were these – there is one short statements (page 6 line 13/14) with „data not shown“. Ultimately single cell RNA profiling would probably be needed to understand the developmental differences/similarities caused by different origins of hypomethylation.

We apologize if the information on the number of RNA-seq replicates was not clear. To account for possible experimental variations, RNA-seq in the initial manuscript was performed on three independent *Dnmt1*^{-/-} and WT littermate embryos, and six independent DKO and control (WT and *Dnmt3a*^{-/+}) littermate embryos. The information on the number of biological replicates can be found in the **Figure 4a** (PCA analysis), **Supplementary Figure 4a-b**, **Supplementary Table 4** and the Methods section. To clarify this point, we also added the following sentence in the text page 6: *"RNA-seq was performed on three *Dnmt1*^{-/-} and three WT littermate embryos, as well as six DKO embryos and six WT and *Dnmt3a*^{-/+} littermate controls (Supplementary Table 4)."*

Moreover the PCA in the **Figure 4a** demonstrates that there is little experimental divergence amongst the control samples and KO samples. We agree that single cell RNA-seq would provide additional insights on the consequences of hypomethylation in development, however we were unable to perform these experiments for this revision.

2) The incredible high correlation of RRBS and WGBS data (Supplementary Figure 1B, C, E) stands in stark contrast to the difference visible in Figure 1B and 1E particularly around the TSS and gene body in WT and in KO's. The overall levels of methylation measured by WGBS differ between 10% and 20% for both CGI's and non CGI's in *Dnmt1*^{-/-} and DKOs (slightly less). Which filtered data are used for the genome wide comparison? I have never seen a correlation of 1 in any type of RRBS or WGBS datasets.

The differences between RRBS and WGBS in **Figure 1b-e** are expected because of the different genome coverage of the two methods. MspI based RRBS is strongly enriched for CpG islands, which explains the low methylation score around the TSS. In contrast the WGBS methylation score around the TSS is higher because it also covers CpG-poor promoters. Despite these global differences, the correlation between RRBS and WGBS is good because it is done only on CpGs covered by both methods. To avoid any confusion, we removed the RRBS/WGBS correlation from the **Supplementary Figure 1**.

As for the high correlation between replicates, we correlate methylation scores averaged in 500bp windows. We apologize if this was not clear and modified the legend of the **Supplementary Figure 1** to make it more explicit. Using this method we routinely obtain correlations of 1 between biological replicates of RRBS in embryos (see for example the Supplementary Figure 3 in Auclair et al., *Genome Res* 2016).

3) The (replicated) RRBS data shown in Figure 1 are mainly used to „validate“ the WGBS data (for which it is unspecified if they are based on replicates as well). Following a first general genome wide comparison (including the RRBS replicates) all subsequent analyses are performed with single (?) WGBS datasets for WT, *Dnmt1*^{-/-}, DKO's hence analysing CpGs at 5x coverage across all samples

– this implies a minimum error range of 20% at individual positions! All other datasets important for interpretations were imported from other (previous) studies.

It is true that RRBS is mainly used to validate the global trends in DNA methylation on a high number of embryos and that most of the subsequent analysis was performed on a single WGBS dataset for each genotype. In our initial WGBS data, 5X was used as a minimal cutoff but the mean sequencing depth was >12X, implying that the vast majority of CpGs do not have a 20% error range. Nevertheless, we generated a second WGBS replicate for each genotype on an independent embryo (**Supplementary Table 2**). WGBS profiles from biological replicates correlate very well (see **Figures 1d, 1g, 2a** and **Supplementary Figures 1e, 1f, 2a**) and confirm all of our initial conclusions. In this revised version, we reanalyzed all the WGBS methylation data by combining biological replicates of each genotype to generate WGBS datasets with a mean sequencing depth around 25X (WT: 24.81X; D1KO: 23.85X, DKO: 25.62X).

4) It remains unclear why *Dnmt1*^{-/-} is not included in comparison to DKO in Figure 2.

We thank the reviewer for this suggestion and included *Dnmt1*^{-/-} tracks in the **Figure 2** and **Supplementary Figure 2**.

5) The experiments shown to support the lack of maintenance contribution of *Dnmt3a/b* in MEFs is unclear. There are some technical concerns: The authors do neither show western blot data or RNA seq data to confirm the timing of *Dnmt3a* and *3b* loss– nor do they comment on effects of long term passaging (do they get possible selection effects when primary MEFs get into crisis). Moreover the experiment does not provide a real information on *Dnmt3a/b* contribution of maintenance. The conclusion is that there is no loss of overall methylation! An induced KO of *Dnmt1* would have been the real control to show to which extent *Dnmt3a/b* contribute to „maintain methylation“.

To further demonstrate the efficient inactivation of *Dnmt3a* and *Dnmt3b* in MEFs, we performed a kinetic analysis by RT-qPCR which confirmed that the expression of the catalytic exons of *Dnmt3a/b* becomes undetectable following Tamoxifen treatment while *Dnmt1* expression is unchanged (**Supplementary Figure 3a**). Furthermore we added a western blot showing that the DNMT3A protein is strongly reduced in Tamoxifen-treated cells compared to untreated cells (**Supplementary Figure 3c**). For DNMT3B we were unable to detect the protein even in untreated MEFs due to the known very low expression of DNMT3B in differentiated cells (data not shown).

We also thank the reviewer for suggesting to use an induced KO of *Dnmt1* as a control. To this end, we derived and immortalized MEFs from a *Dnmt1* 2lox/2lox embryo and infected them with a virus coding for the tamoxifen-inducible CRE recombinase Cre-ERT2 to perform conditional inactivation of *Dnmt1* (**Supplementary Figure 3g**). Tamoxifen treatment induced rapid and efficient recombination of the *Dnmt1* 2lox alleles (**Supplementary Figure 3h-i**). Furthermore, RRBS experiments on three independent inductions showed that the induced KO of *Dnmt1* leads to an immediate and global hypomethylation of genomic DNA (**Supplementary Figure 3j-k**), strongly suggesting that DNMT1 alone is necessary and sufficient for maintenance methylation. Notably, in contrast to *Dnmt3a/b* cDKO MEFs, *Dnmt1* cKO MEFs rapidly stopped dividing,

All these new results are presented in the text page 6. Altogether, these new experiments presented in the **Supplementary Figure 3** reinforce our conclusion of a lack of maintenance activity of *Dnmt3a/b* in MEFs.

Minor:

The authors do not explain how RRBS libraries were constructed (by MspI digestion?) and if this explains (biased) differences between WGBS and RRBS.

We clarified in page 3 and the Methods section that the RRBS libraries were constructed by MspI digestion, which indeed explains the differences between RRBS and WGBS because RRBS is biased towards CpG islands.

The reason why the author speak about „heterochromatic“ clusters in the text referring to Suppl Figure 8D remains totally unclear. There are no data showing the heterochromatin connection. We agree and removed the word "heterochromatic".

The nice examples of the sashimi plots in supple figure 10 e.g. for Cyp2b23 should be included into the main figure 6.

We thank the reviewer for this suggestion and reorganized the **Figure 5** to include the RNA-seq tracks and sashimi plots for *Cyp2b23*.

The authors state in their discussion that the results provide „an important clarification on the function of“ DNMT's – most of functional aspects have been known /discussed before and are „comprehensively validated“ but certainly not clarified.

We agree and changed the discussion to say that our results provide a "comprehensive validation of the function" of DNMTs *in vivo*.

Reviewer #3 (Remarks to the Author):

For about two decades, it has been known that DNA methyltransferases, DNMT1, DNMT3A, and DNMT3b, are essential for mouse embryonic development. In this manuscript, Dahlet et al. report the first genome-wide methylation analyses on Dnmt1 KO and Dnmt3ab DKO embryos, which revealed that Dnmt3a and Dnmt3b are together required for de novo methylation after implantation whereas Dnmt1 is solely required for maintenance methylation. They also revealed that DNA methylation is important to repress a number of germline genes as well as many retrotransposons such as IAP, L1, and MERVK10C. Their findings also include that DNA methylation plays a role in suppressing cryptic transcription in gene bodies, and that CpG-rich promoters of lineage-committed genes once gain DNA methylation after the genome-wide erasure of methylation in ICM, and this methylation is important to repress these genes in unrelated tissues. The manuscript is generally well written and data are presented appropriately.

However, most of the results seem to be an extension of the previous knowledge (reported before the NGS era), which declines the conceptual novelty of the manuscript. I have some concerns listed below.

(1) For expression analyses, RNAs were prepared by ribosome depletion (Ribo-zero). Thus, pre-mRNAs were included in the RNA preparation. Because many TEs reside in introns, polyA(+) RNA purification is better to quantify the TE expression. Alternatively, the authors should provide some data supporting that the existence of pre-mRNAs does not have a significant impact.

We agree that this is a possible concern and that TEs could be called upregulated because they are located in an intron of an upregulated gene. To demonstrate that pre-mRNAs have little impact on TE upregulation in the mutant embryos, we generated a new figure (**Supplementary Figure 9e**) showing that only 2% of the upregulated TEs in *Dnmt1*^{-/-} embryos are located in an intron of an upregulated gene. Moreover, in these cases, gene activation can be a consequence of TE activation rather than the opposite case (as shown in **Figure 5f**). Thus it is unlikely that the list of upregulated TEs in KO embryos is biased by the Ribo-zero library preparation.

(2) To identify unregulated genes by RNA-seq, the sequence reads that were mapped on to genomic regions annotated as a TE should be removed from the bam files after Tophat mapping, because they could be incorrectly mapped onto TE sequences present in UTRs of genes and this could give false upregulation if the same family of TE copies somewhere in the genome are upregulated.

We thank the reviewer for this very pertinent suggestion. We reanalyzed differential gene expression from the RNA-seq datasets after removing the reads mapped to TEs from the bam files. The log₂FC of most genes - in particular upregulated genes - are very similar compared to the old analysis (see for example *Dnmt1*^{-/-} embryos in the figure below, left panel). Nevertheless, the reviewer was right in the sense that this highlighted a small number of genes that were falsely called upregulated because of the presence of an upregulated TE in the 3'UTR of the gene, as exemplified by the *Zfp575* gene in the figure below (right panel). These genes are no longer called upregulated in the new analysis.

All the figures and supplementary data in the revised manuscript have been updated with the new lists and results of differential gene expression analysis.

(3) In page 9, lines 8 to 18 (results shown in Fig. 5B), the authors counted upregulated TE copies in the genome by using uniquely mapped reads. It should be mentioned that this method underestimates it, because very young TE copies (which likely retain the transcriptional potential) cannot be uniquely mapped. Less than 0.1% of L1Md_A copies showed upregulation. But when analyzed in bulk as shown in Fig. S9C, L1Md_A showed about 4-fold upregulation. Same for L1Md_Tf.

We thank the reviewer for this suggestion and added the following sentence in the text page 9: *"It should be mentioned that this method underestimates the counts of upregulated TE copies because very young TEs cannot be uniquely mapped"*.

(4) In page 9, middle, the authors identified TEs that drive expression of nearby genes, similar to the situation of *Setdb1* KO ES cells (ref.11). It should also be mentioned that loss of DNA methylation and TE upregulation leads to the upregulation of nearby genes in spermatogenic cells (Inoue et al. PLoS Genet 2017, 13: e1006926).

We thank the reviewer for pointing out this important reference. We added the following sentence in the text page 10:

*"Altogether this shows that derepressed TEs alter the expression of nearby genes, similarly to previous observations made in *Setdb1* KO ES cells (ref 23) and upon loss of DNA methylation in *Dnmt3l* mutant spermatocytes (ref 42)"*.

(5) In page 9, lines 25 to 27, it should be mentioned that the IAP sequences are inserted in the antisense orientation to the host genes, *Trpm2* and *Capn11*. When analyzing TE expression using the rebase sequences, did the authors see upregulation of the antisense transcription of IAPLTR1_Mm?

We modified the text page 10 as follows:

"We also observed intragenic initiation from intronic IAPeZ and its flanking LTR IAPLTR1_Mm inserted in the antisense orientation to the host gene, as exemplified by the Capn11, Trpm2, and Apoh genes."

Furthermore, to investigate if antisense transcription from IAPLTR1_Mm is frequent, we generated heatmaps and metaplots of RNA-seq signal in sense and antisense orientation from all IAPLTR1_Mm elements (**Supplementary Figure 10e-f**). This analysis confirmed that there is a noticeable upregulation of antisense transcription from several IAPLTR1_Mm elements in *Dnmt1*^{-/-} embryos. We added the following sentence in the text page 10:

"To determine if antisense transcription from IAPLTR1_Mm elements is frequent, we counted the RNA signal from all IAPLTR1_Mm elements and found that, while there is a strong activation in the sense orientation, there is also a noticeable increase in antisense transcription initiation from these elements in Dnmt1^{-/-} embryos (Supplementary Figure 10e-f)."

(6) In the previous report (Inoue et al. PLoS Genet 2017), loss of methylation upregulates the antisense L1 promoters. This reviewer wonders if this was also observed in the *Dnmt1* KO and *Dnmt3ab* DKO embryos.

We thank again the reviewer for this question. Interestingly, in contrast to Inoue et al. we do not see upregulation of genes by nearby L1 retrotransposons in *Dnmt* mutant embryos. We added the following sentence in the text page 10:

"Interestingly, in contrast to Dnmt3L KO spermatocytes (ref 42), we did not see activation of nearby genes by L1 retrotransposons, suggesting that the impact of hypomethylated TEs on the integrity of the transcriptome is different between germ cells and somatic cells."

(7) Related to the section in page 8, "de novo methylation is required to repress 2C-specific genes", it is possible that the presence of DNMT3A and/or DNMT3B proteins is required for the repression. For example, they may bind and recruit a silencing complex to these promoters. This can be tested by using the catalytically inactive mutants, which can be left for future studies. But, at least, it is better to discuss this possibility.

We agree with the reviewer that our results do not support a direct role of DNA methylation in repressing 2C genes but most likely reflect accessory functions of DNMT3A/B. Therefore we changed the title of the paragraph from "*De novo DNA methylation is required to repress 2C-specific genes*" to "*De novo DNA methyltransferases are required to repress 2C-specific genes*".

Furthermore, we expanded the discussion to insist on possible indirect roles of DNA methylation or non-catalytic functions of DNMT3A/B enzymes. The following text was added in the discussion:

"Adding to the complexity, Dux and 2C genes are not strongly activated despite hypomethylation in Dnmt1^{-/-} embryos, suggesting either an indirect regulation by DNA methylation or a possible role of non-catalytic functions of DNMT3A/B. Hence, it is possible that DNMT3A/B repress 2C genes by recruiting silencing complexes to these promoters independently of their catalytic activity (ref 61,62)."

(8) In page 10, lines 13-15, the authors stated the sites of cryptic transcription start are methylated in WT embryos. Did they see loss of methylation in the *Dnmt* mutants? The statistical data for the mutants should be presented in Fig. 6C along with the WT data. Related to this, whereas only the

example of Mgl2 is shown in Fig. 6AD, a few other examples can be included as supplementary figures to demonstrate the generality.

As requested, we included DNA methylation of *Dnmt* mutants in the **Figure 6c**, which confirms that the sites of cryptic initiation lose DNA methylation in *Dnmt* mutants. Furthermore, three more examples of genes with cryptic transcription initiation are shown in the **Supplementary Figure 11**.

(9) In page 4, line 11, "...occurs uniformly across all genomic sequences...". As far as inspected for Fig. 4G (the *Dnmt1*^{-/-} track) and Fig. 11, it does not look uniform. Regions of >80% methylation in wild-type showed 10-50% methylation remained in the *Dnmt1* KO. Related to this, it is better to include a *Dnmt1* KO track in Fig. 2A to show genomic regions with high methylation levels in E3.5 ICM and E8.5 DKO indeed lost methylation in E8.5 *Dnmt1* KO.

We agree with the reviewer that the word "uniformly" was inappropriate and we removed it from the text. Indeed the patterns of DNA methylation in *Dnmt1* KO embryos are not strictly uniform, which probably reflects varying efficacies of continuous *de novo* methylation activity.

As requested, we now provide several *Dnmt1* KO WGBS tracks in the **Figure 2** and **Supplemental Figure 2** showing that regions with high methylation in E3.5 ICM and E8.5 DKO indeed lost methylation in *Dnmt1* KO embryos.

(10) Page 4 line 34 and Page 6, line 19. The authors stated "strong positive correlation" and "good correlation" respectively. Although these correlations can be seen in Fig. 2B and Fig. S4D, the R values should be provided.

The r values have been included in the **Figure 2b** ($r = 0.80$) and the **Supplementary Figure 4d** ($r = 0.66$).

(11) In Fig. H, it is unclear from the main text and legend whether single CpG sites were analyzed or single bisulfite reads were analyzed.

We apologize if this was not clear. We modified the legend of **Figure 1h** to clarify that the figure depicts CpG methylation averaged in 1kb windows.

(12) In Fig. S5AC, please indicate the methylation levels (%). In Figs. 4E and S5E, please indicate the p-values, as the *Dazl* expression with gRNA showed a large experimental variation.

The methylation levels (%) are now indicated next to the bisulfite sequencing results in the **Supplementary Figure 5**. In addition, we added p-values in the **Figure 4e** and **Supplementary Figure 5e**, which confirmed the significance of the observed differences including for the *Dazl* gene.

REVIEWERS' COMMENTS:

Reviewer #1 (Remarks to the Author):

Thank you for addressing all my comments satisfactorily.

Reviewer #3 (Remarks to the Author):

In the revised manuscript, the authors have addressed all the concerns I had. Although most of the results seem to be an extension of the previous knowledge, as I stated in the earlier review, the current version of the manuscript with the comprehensive genome-wide analyses is worth publishing for advancing the research field.

Kenji Ichiyanagi